# Flow-Based Policy for Online Reinforcement Learning

**Lei Lv**[*1,2,3,]**, Yunfei Li**[2,3]**, Yu Luo**[3]**, Fuchun Sun**[†3]**, Tao Kong**[2]**, Jiafeng Xu**[†2]**, Xiao Ma**[2]

[1] Shanghai Research Institute for Intelligent Autonomous Systems, Tongji University
[2] ByteDance Seed
[3] Tsinghua University

```
2311830@tongji.edu.cn;{liyunfei.cloud, taokong, xjf, xiao.ma}@bytedance.com;
                    {luoyu19,fcsun}@tsinghua.edu.cn
```

## Abstract

We present **FlowRL**, a novel framework for online reinforcement learning that integrates flow-based policy representation with Wasserstein-2-regularized optimization. We argue that in addition to training signals, enhancing the expressiveness of the policy class is crucial for the performance gains in RL. Flow-based generative models offer such potential, excelling at capturing complex, multimodal action distributions. However, their direct application in online RL is challenging due to a fundamental objective mismatch: standard flow training optimizes for static data imitation, while RL requires value-based policy optimization through a dynamic buffer, leading to difficult optimization landscapes. FlowRL first models policies via a state-dependent velocity field, generating actions through deterministic ODE integration from noise. We derive a constrained policy search objective that jointly maximizes Q through the flow policy while bounding the Wasserstein-2 distance to a behavior-optimal policy implicitly derived from the replay buffer. This formulation effectively aligns the flow optimization with the RL objective, enabling efficient and value-aware policy learning despite the complexity of the policy class. Empirical evaluations on DMControl and Humanoidbench demonstrate that FlowRL achieves competitive performance in online reinforcement learning benchmarks.We have released our code here.

## 1 Introduction

Recent advances in iterative generative models, particularly Diffusion Models (DM) [16, 38] and Flow Matching (FM) [23, 24, 41], have demonstrated remarkable success in capturing complex multimodal distributions. These models excel in tasks such as high-resolution image synthesis [7], robotic imitation learning [6, 2], and protein structure prediction [18, 3], owing to their expressivity and ability to model stochasticity. A promising yet underexplored application lies in leveraging their multimodal generation capabilities to enhance reinforcement learning (RL) policies, particularly in environments with highly stochastic or multimodal dynamics.

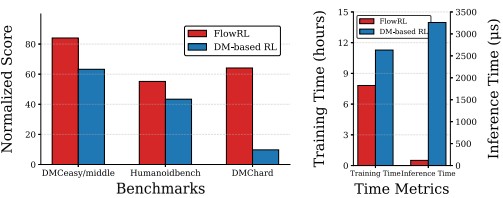

Figure 1: (left) Normalized scores comparing FlowRL and DM-based RL (QVPO) on 12 challenging DMC-hard and HumanoidBench tasks, and 3 DMC-easy & middle tasks. (right) Computational efficiency on the Dogrun task: 1M-step training time and single env step inference time.

---

[*]The work was accomplished during the authors' internship at ByteDance Seed.
[†]Corresponding author.

Traditional RL frameworks alternate between Q-function estimation and policy updates [39], often parameterizing policies as Gaussian [13] or deterministic policies [37, 12] to maximize expected returns. However, directly employing diffusion or flow-based models as policies introduces a fundamental challenge: the misalignment between RL objectives, which aim to optimize value-aware distributions, and generative modeling, which imitates static data distributions. This discrepancy becomes exacerbated in online RL, where nonstationary data distributions and evolving Q-value estimates lead to unstable training [12].

While recent methods have pioneered the use of diffusion model (DM)-based policies in online reinforcement learning [45, 8, 43], these approaches still suffer from high computational cost and inefficient sample usage (see Section 2). By contrast, flow-based models (FMs), despite their ability to represent complex and multimodal policies, have yet to be effectively integrated into online RL frameworks.

Our method distinguishes itself by leveraging carefully selected replay buffer data as a reference distribution to align flow-based policies with high-value behaviors while preserving multimodality. Inspired by prior works such as SIL [28] and OBAC [26], which utilised behaviour policies to guide policy optimization but limit policy expressivity to capture diverse behaviors, we propose a unified framework that integrates flow-based action generation with Wasserstein-2-regularized [10] distribution matching. Specifically, our policy extraction objective simultaneously maximizes Q-values through flow-based actor and minimizes distribution distance from high-reward trajectories identified in the replay buffer. By reformulating this dual objective as a guided flow-matching loss, we enable the policy to adaptively imitate empirically optimal behaviors while exploring novel actions that maximize future returns. Besides, this approach retains the simplicity of standard actor-critic architectures, without requiring lengthy iterative sampling steps or auxiliary inference tricks [19, 8]—yet fully exploits the multimodality of flow models to discover diverse, high-performing policies. We evaluate our approach on challenging DMControl [40] and HumanoidBench [36], demonstrating competitive performance against state-of-the-art baselines. Notably, our framework achieves one-step policy inference, significantly reducing computational overhead and training instability caused by backpropagation through time (BPTT) [43, 31]. Experimental results highlight both the empirical effectiveness of our method and its practical advantages in scalability and efficiency, establishing a robust pathway for integrating expressive generative models into online RL.

## 2  Related Work

In this section, we provide a comprehensive survey of existing policy extraction paradigms based on iterative generative models based policy, with a particular focus on recent advances that leverage diffusion and flow-based models in offline or online reinforcement learning. We categorize these approaches according to their underlying policy optimization objective and highlight their respective advantages and limitations.

**Generalized Behavior Cloning**    Generalized Behavior Cloning, often akin to weighted behavioral cloning or weighted regression [33, 32], trains policies by imitating high-reward trajectories from a replay buffer, weighted by advantage or value estimates, thereby avoiding BPTT. Previous methods like EDP [19], QGPO [25], QVPO [8], and QIPO [46] implemented these paradigms, enhancing computational efficiency by bypassing BPTT. However, as demonstrated in prior research, this approach has been empirically shown to be inefficient [30, 31], and often leads to suboptimal performance.

**Reverse process as policy parametrizations**    These methods use reparameterized policy gradients, computing gradients of the Q-function with respect to policy parameters directly through the generative model's reverse sampling process, similar to the reparameterization trick commonly employed in Gaussian-based policies [13]. Previous methods, such as DQL [44], DiffCPS [15], Consistency-AC [9], and DACER [43], backpropagate gradients through the reverse diffusion process, which, while flexible, incurs significant computational costs due to iterative denoising and backpropagation through time (BPTT) [30]. These factors limit the scalability of such algorithms to more complex environments. To address this, FQL [31] distills a one-step policy from a flow-matching policy, reducing computational cost, but requires careful hyperparameter tuning.

**Other Approaches.**   Beyond above methods, alternative methods include action gradients [45, 34], hybrid Markov Decision Processes (MDPs) [35] , rejection sampling [4] or combinations of above strategies [27].

The distinction between these methods underscores an inherent trade-off between computational simplicity and the efficiency of policy extraction. Generalized Behavior Cloning emphasizes ease of implementation, often at the expense of policy extraction efficiency. In contrast, reparameterized policy gradients facilitate direct policy updates but incur increased complexity. These observations highlight the necessity for further research to achieve a better balance between expressivity and scalability when applying iterative generative models to reinforcement learning.

## 3 Preliminaries

### 3.1 Reinforcement Learning

Consider the Markov Decision Process (MDPs) [1] defined by a 5-tuple $\mathcal{M} = \langle \mathcal{S}, \mathcal{A}, \mathcal{P}, r, \gamma \rangle$, where $\mathcal{S} \in \mathbb{R}^n$ and $\mathcal{A} \in \mathbb{R}^m$ represent the continuous state and action spaces, $\mathcal{P}(s'|s, a) : \mathcal{S} \times \mathcal{A} \to \Delta(\mathcal{S})$ denotes the dynamics distribution of the MDPs, $r(s, a) : \mathcal{S} \times \mathcal{A} \to \Delta(\mathbb{R})$ is a reward function, $\gamma \in [0, 1)$ gives the discounted factor for future rewards. The goal of RL is to find a policy $\pi(a|s) : \mathcal{S} \to \Delta(\mathcal{A})$ that maximizes the cumulative discounted reward:

$$J_\pi = \mathbb{E}_{\pi, \mathcal{P}} \left[ \sum_{t=0}^{\infty} \gamma^t r(s_t, a_t) \right]. \tag{1}$$

In this paper, we focus on the online off-policy RL setting, where the agent interacts with the environment and collects new data into a replay buffer $\mathcal{D} \leftarrow \mathcal{D} \cup \{(s, a, s', r)\}$. The replay buffer consequently maintains a distribution over trajectories induced by a mixture of historical behavior policies $\pi_\beta$. At the $k$-th iteration step, the online learning policy is denoted as $\pi_k$, with its corresponding $Q$ value function defined by:

$$Q^{\pi_k}(s, a) = \mathbb{E}_{\pi_k, \mathcal{P}} \left[ \sum_{t=0}^{\infty} \gamma^t r(s_t, a_t) | s_0 = s, a_0 = a \right], \tag{2}$$

and it can be derived by minimizing the TD error [39]:

$$\arg\min_{Q^{\pi_k}} \mathbb{E}_{(s,a,r,s') \sim \mathcal{D}} \left[ (Q^{\pi_k}(s, a) - \mathcal{T}^{\pi_k} Q^{\pi_k}(s, a))^2 \right],$$
$$\text{where} \quad \mathcal{T}^{\pi_k} Q^{\pi_k}(s, a) = r(s, a) + \gamma \mathbb{E}_{s' \sim \mathcal{P}(\cdot|s,a), \, a' \sim \pi_k(\cdot|s')} \left[ Q^{\pi_k}(s', a') \right]. \tag{3}$$

Similarly, we distinguish the following key elements:

- **Optimal policy and Q-function:** The optimal policy $\pi^*$ maximizes the expected cumulative reward, and the associated Q-function $Q^*(s, a)$ characterizes the highest achievable return.

- **Behavior policy and replay buffer:** The behavior policy $\pi_\beta$ is responsible for generating the data stored in the replay buffer [22, 26]. Its Q-function, $Q^{\pi_\beta}(s, a)$, reflects the expected return when following $\pi_\beta$. Notably, $\mathcal{D}$ is closely tied to the distribution of $\pi_\beta$, such that actions sampled from $\mathcal{D}$ are supported by those sampled from $\pi_\beta$ (i.e., $a \in \mathcal{D} \Rightarrow a \sim \pi_\beta$).

- **Behavior-optimal policy:** Among all behavior policies present in the buffer, we define $\pi_{\beta^*}$ as the one that achieves the highest expected return, with Q-function $Q^{\pi_{\beta^*}}(s, a)$.

These definitions yield the following relationship, which holds for any state-action pair:

$$Q^*(s, a) \geq Q^{\pi_{\beta^*}}(s, a) \geq Q^{\pi_\beta}(s, a). \tag{4}$$

This relationship suggests that, although direct access to the optimal policy is typically infeasible, the value of the optimal behavior policy constitutes a theoretical lower bound [28] on the performance that can be achieved by policies derived from the replay buffer.

## 3.2 Flow Models

Continuous Normalizing Flows (CNF) [5] model the time-varying probability paths by defining a transformation between an initial distribution $p_0$ and a target data distribution $p_1$ [23, 24]. This transformation is parameterized by a flow $\psi_t(x)$ governed by a learned time-dependent vector field $v_t(x)$ [5], following the ordinary differential equation (ODE):

$$\frac{d}{dt}\psi_t(x) = v_t(\psi_t(x)), \tag{5}$$

and the continuity equation [42]:

$$\frac{d}{dt}p_t(x) + \nabla \cdot [p_t(x)v_t(x)] = 0, \quad \forall x \in \mathbb{R}^d. \tag{6}$$

**Flow Matching.** Flow matching provides a theoretically grounded framework for training continuous-time generative models through deterministic ordinary differential equations (ODEs). Unlike diffusion models that rely on stochastic dynamics governed by stochastic differential equations (SDEs) [38], flow matching operates via a *deterministic* vector field, enabling simpler training objectives and more efficient sampling trajectories. The core objective is to learn a neural velocity field $v_\theta : [0, 1] \times \mathbb{R}^d \rightarrow \mathbb{R}^d$ that approximates a predefined conditional target velocity field $u(t, x|x^1)$.

Given a source distribution $q(x^0)$ and target distribution $p(x^1)$, the training process involves minimizing the conditional flow matching objective [23]:

$$\mathcal{L}_{\text{CFM}}(\theta) = \mathbb{E}_{\substack{t \sim \mathcal{U}([0,1]) \\ x^1 \sim p, \, x^0 \sim q}} \left\| v_\theta(t, x^t) - u(t, x^t|x^1) \right\|_2^2, \tag{7}$$

where the linear interpolation path is defined as $x^t = tx^1 + (1 - t)x^0$ with $u(t, x^t|x^1) = x^1 - x^0$. This formulation induces a *probability flow* governed by the ODE:

$$\frac{dx}{dt} = v_\theta(t, x), \quad x^0 \sim q, \tag{8}$$

which transports samples from $q$ to $p$.

# 4 Method

In this section, we detail the design of our method. We first parameterize the policy as a flow model, where actions are generated by integrating a learned velocity field over time. For policy improvement, we model policy learning as a constrained policy search that maximizes expected returns while bounding the distance to an optimal behavior policy. Practically, we circumvent intractable distribution matching and optimal behavior policy by aligning velocity fields with elite historical actions through regularization and implicit guidance, enabling efficient constraint enforcement.

## 4.1 Flow Model based Policy Representation.

We parameterize $\pi_\theta$ with $v_\theta(t, s, a^t)$, a state-action-time dependent velocity field, as an actor for reinforcement learning. The policy $\pi_\theta$ can be derived by solving ODE (8) :

$$\pi_\theta(s, a^0) = a^0 + \int_0^1 v_\theta(t, s, a^t)dt, \tag{9}$$

where $a^0 \sim \mathcal{N}(0, I^2)$. The superscript $t$ denotes the continuous time variable in the flow-based ODE process to distinguish it from discrete Markovian time steps in reinforcement learning. (For brevity, the terminal condition at $t = 1$ is omitted in the notation.) The Flow Model derives a deterministic velocity field $v_\theta$ from an ordinary differential equation (ODE). However, when $a^0$ is sampled from a random distribution, the model effectively functions as a stochastic actor, exhibiting diverse behaviors across sampling instances. This diversity in generated trajectories inherently promotes enhanced exploration in online reinforcement learning.

Recall the definition in Section 3.1. Following the notation of $\pi_\beta$ and $\pi_{\beta*}$, we can define the corresponding velocity fields as follows:

Let $v_\beta$ be the velocity field induced by the behavior policy $\pi_\beta$, such that:

$$v_\beta(s, a) = a - a^0.$$

where $s, a \sim \mathcal{D}$, and $a^0 \sim \mathcal{N}(0, I^2)$.

Similarly, let $v_{\beta^*}$ denote the velocity field induced by the behavior-optimal policy $\pi_{\beta^*}$:

$$v_{\beta^*}(s, a) = a - a^0.$$

where $a \sim \pi_{\beta^*}$, and $a^0 \sim \mathcal{N}(0, I^2)$.

## 4.2 Optimal-Behavior Constrained Policy Search with Flow Models

Building on the discussion in Section 3.1, where the optimal behavior policy is established as a lower bound for the optimal policy, we proceed to optimize the following objective under a constrained policy search setting:

$$\theta^* = \arg\max_\theta \ \mathbb{E}_{a \sim \pi_\theta}\left[ Q^{\pi_\theta}(s, a) \right],$$
$$\text{s.t. } D\left( \pi_\theta, \pi_{\beta^*} \right) \leq \epsilon. \tag{10}$$

Here, $D(\pi_\theta, \pi_{\beta^*})$ denotes a distance metric between the current policy and the optimal behavior policy distributions.

The objective is to maximize the expected reward $\mathbb{E}_{a \sim \pi_\theta}[Q^\pi(s, a)]$ while constraining the learned policy $\pi_\theta$ to remain within an $\epsilon$-neighborhood of the optimal behavior policy $\pi_{\beta^*}$, i.e., $D(\pi_\theta, \pi_{\beta^*}) \leq \epsilon$. This formulation utilizes the Q-function, a widely used and effective approach for policy extraction, while ensuring fidelity to the optimal behavior policy.

Despite its theoretical appeal, this optimization paradigm exhibits two inherent limitations:

- *Challenges in computing distributional distances*: For flow-based models, computing policy densities at arbitrary samples is computationally expensive, which limits the practicality of distance metrics such as the KL divergence for sample-based estimation and policy regularization.

- *Inaccessibility of the optimal behavior policy $\pi_{\beta^*}$*: The replay buffer contains trajectories from a mixture of policies, making it difficult to directly sample from $\pi_{\beta^*}$ or to reliably estimate its associated velocity field, thereby complicating the computation of related quantities in practice.

## 4.3 A Tractable Surrogate Objective

To overcome the aforementioned challenges, we propose the following solutions:

- **Wasserstein Distance as Policy Constraints:** We introduce a policy regularization method based on the alignment of velocity fields. This approach bounds the Wasserstein distance between policies by characterizing their induced dynamic transport processes, thereby imposing direct empirical constraints on the evolution of policies without requiring density estimation.

- **Implicit Guidance for Optimal Behaviors:** Instead of explicitly constraining the policy to match the inaccessible $\pi_{\beta^*}$, we leverage implicit guidance from past best-performing behaviors in the buffer, enabling efficient revisiting of arbitrary samples and encouraging the policy to remain within a high-quality region of the action space.

In particular, we adopt the squared Wasserstein-2 distance for its convexity with respect to the policy distribution and ease of implementation. This metric is also well-suited for measuring the velocity field between policies and enables efficient sample-based regularization within the flow-based modeling framework. In general, we can define the Wasserstein-2 Distance [42] as follows :

**Definition 4.1 (Wasserstein-2 Distance)** Given two probability measures $p$ and $q$ on $\mathbb{R}^n$, the squared Wasserstein-2 distance between $p$ and $q$ is defined as:

$$W_2^2(p, q) = \inf_{\gamma \in \Pi(p, q)} \int_{\mathbb{R}^n \times \mathbb{R}^n} \gamma(x, y) \|x - y\|^2 dx dy, \tag{11}$$

where $\Pi(p, q)$ denotes the joint distributions of $p$ and $q$, $\gamma$ on $\mathbb{R}^n \times \mathbb{R}^n$ with marginals $p$ and $q$. Specifically, we derive a tractable upper bound for the Wasserstein-2 distance (proof in A.1):

**Theorem 4.1 (W-2 Bound for Flow Matching)** Let $v_\theta$ and $v_{\beta*}$ be two velocity fields inducing time-evolving distributions $\pi_\theta^t(a|s)$ and $\pi_{\beta*}^t(a|s)$, respectively. Assume $v_\beta$ is Lipschitz continuous in $a$ with constant $L$. $a^t = ta + (1-t)a^0$. Then, the squared Wasserstein-2 distance between $\pi_\theta$ and $\pi_{\beta*}$ at $t = 1$ satisfies:

$$W_2^2\left(\pi_\theta, \pi_{\beta*}\right) \leq e^{2L} \int_0^1 \mathbb{E}_{a \sim \pi_{\beta*}} \left[\left\|v_\theta(s, a^t, t) - v_{\beta*}\right\|^2\right] dt. \tag{12}$$

By explicitly constraining the Wasserstein-2 distance, the model enforces proximity between the current policy and the optimal policy stored in the buffer. This objective is inherently consistent with the generative modeling goal of minimizing distributional divergence. The regularization mechanism benefits from the representational expressiveness of flow-based models in capturing diverse, high-performing action distributions while systematically restricting policy updates.

However, while the upper bound of Wasserstein-2 distance above is theoretically tractable, sampling directly from $\pi_{\beta*}$ or evaluating its velocity field remains a computational barrier in practice. To circumvent this limitation, we introduce an implicit guidance (13) mechanism through the $Q^{\pi_{\beta*}}$, which is more readily estimable:

$$\mathbb{E}_{\substack{a' \sim \pi_\theta, t \sim \mathcal{U}(0,1) \\ s,a \sim \mathcal{D}}} \left[f\left(Q^{\pi_{\beta*}}(s,a) - Q^{\pi_\theta}(s,a')\right) \left\|v_\theta(s, a^t, t) - (a - a^0)\right\|^2\right], \tag{13}$$

$$f \propto \max\left(Q^{\pi_{\beta*}} - Q^{\pi_\theta}, 0\right). \tag{14}$$

The constraint incorporates a non-negative weighting function, as defined in Eq. (14), thereby establishing an adaptive regularization mechanism. A positive value of $f$ signifies that the behavioral policy achieves superior performance relative to the current policy; under these circumstances, the constraint adaptively regularizes the current policy towards the optimal behavioral policy.

The implicit form of the constraints in Eq. (12) enables efficient utilization of arbitrary samples from the replay buffer, thus improving sample efficiency. Moreover, by relaxing the strict constraint on the Wasserstein-2 distance, the modified objective enhances computational efficiency. Notwithstanding this relaxation, policy improvement guarantees remain valid, as demonstrated in the following theorem (proof in Appendix A.2):

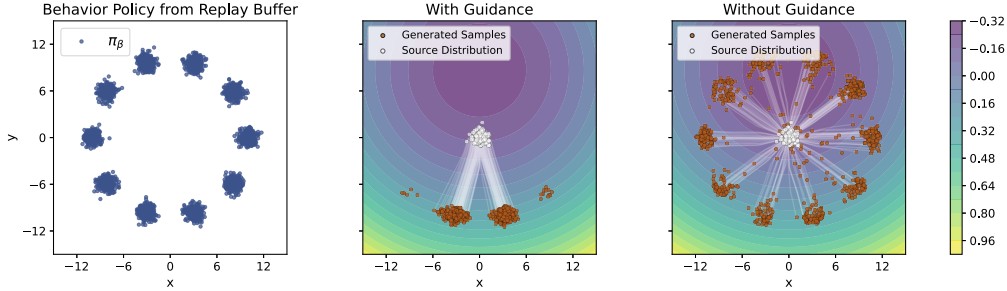

Figure 2: Illustration of Theorem 4.2 on a bandit toy example: (left) behavior data in the replay buffer; (middle) implicit value-guided flow matching steers the policy toward the high-performance behavior policy($\pi_{\beta*}$), heatmap shows $Q^{\pi_k} - Q^{\pi_{\beta*}}$, white lines indicate transport paths; (right) standard flow matching leads to dispersed sampling with high variance under limited flow steps.

**Theorem 4.2 (Weighted CFM)** Let $\pi_k(a|s)$ be the current policy induced by velocity field $v_{\theta_k}$, and $f$, a non-negative weighting function with $f \propto Q^{\pi_{\beta*}} - Q^{\pi_k}$. Minimizing the objective (13) yields an improved policy distribution:

$$\pi_{k+1}(a|s) = \frac{f(s,a)\pi_{\beta*}(a|s)}{\mathcal{Z}(s)}, \tag{15}$$

where $\mathcal{Z}(s) = \int_\mathcal{A} f \cdot \pi_k(a|s)\, da$ is the normalization factor.

Figure 23 shows that, as guaranteed by Theorem 4.2, flow matching with guidance can steer the policy toward the $\pi_{\beta^*}$, even without direct sampling from it. For details of the toy example settings, see Appendix B.4.

## 4.4 A Practical Implementation

Building on the theoretical developments above, we now present a practical implementation of FlowRL, as detailed in Algorithm 1.

**Policy Evaluation** Recall the constraint in Eq. (13), which necessitates the evaluation of both the current policy value function $Q^{\pi_\theta}$ and the optimal behavioral policy value function $Q^{\pi_{\beta^*}}$. The value function $Q^{\pi_\theta}$ is estimated using standard Bellman residual minimization, as described in Eq. (3). For $Q^{\pi_{\beta^*}}$, leveraging the definition of $\pi_{\beta^*}$, we similarly adopt the following objective:

$$\arg \min_{Q^{\pi_{\beta^*}}} \ \mathbb{E}_{(s,a,r,s')\sim\mathcal{D}} \left[ (Q^{\pi_{\beta^*}}(s,a) - \mathcal{T}^{\pi_{\beta^*}} Q^{\pi_{\beta^*}}(s,a))^2 \right], \tag{16}$$

$$\mathcal{T}^{\pi_{\beta^*}} Q^{\pi_{\beta^*}}(s,a) = r(s,a) + \gamma \, \mathbb{E}_{s'\sim\mathcal{D}} \left[ \max_{a'\sim\mathcal{D}} Q^{\pi_{\beta^*}}(s',a') \right]. \tag{17}$$

To circumvent the difficulties of directly evaluating the max operator, we leverage techniques from offline reinforcement learning to estimate $Q^{\pi_{\beta^*}}$. Among these approaches, we adopt expectile regression [20] due to its simplicity and compatibility with unmodified data pipelines. Specifically, the value function $V^{\pi_{\beta^*}}$ and the action-value function $Q^{\pi_{\beta^*}}$ are estimated by solving the following optimization problems:

$$\arg \min_{V^{\pi_{\beta^*}}} \mathbb{E}_{(s,a)\sim\mathcal{D}} \left[ L_2^\tau \left( Q^{\pi_{\beta^*}}(s,a) - V^{\pi_{\beta^*}}(s) \right) \right], \tag{18}$$

$$\arg \min_{Q^{\pi_{\beta^*}}} \mathbb{E}_{(s,a,s',r)\sim\mathcal{D}} \left[ \left( r + \gamma V^{\pi_{\beta^*}}(s') - Q^{\pi_{\beta^*}}(s,a) \right)^2 \right], \tag{19}$$

where $L_2^\tau(x) = |\tau - \mathbb{1}(x < 0)|x^2$ denotes the expectile regression loss and $\tau$ is the expectile factor.

**Policy Extraction** Accordingly, the policy extraction problem for flow-based models can be formulated as the following constrained optimization:

$$\theta^* = \arg \max_\theta \ \mathbb{E}_{s\sim\mathcal{D}, a\sim\pi_\theta} \left[ Q^{\pi_\theta}(s,a) \right], \tag{20}$$

$$\text{s.t.} \ \mathbb{E}_{s,a\sim\mathcal{D}, a'\sim\pi_\theta} \left[ f\left( Q^{\pi_{\beta^*}} - Q^{\pi_\theta} \right) \left\| v_\theta(s, a^t, t) - (a - a^0) \right\|^2 \right] \leq \epsilon. \tag{21}$$

Although a closed-form solution can be derived using the Lagrangian multiplier and KKT conditions, it is generally intractable to apply in practice due to the unknown partition function [32, 33, 26]. Therefore, we adopt a Lagrangian form, leading to the following objective:

$$\mathcal{L}(\theta) = \mathbb{E}_{s,a\sim\mathcal{D}, a'\sim\pi_\theta} [\underbrace{Q^{\pi_\theta}(s,a')}_{\text{exploration}} - \lambda \left( \underbrace{f(Q^{\pi_{\beta^*}} - Q^{\pi_\theta})\|v_\theta - (a - a^0)\|^2}_{\text{exploitation}} - \epsilon \right)]. \tag{22}$$

Where $\lambda$ is the Lagrangian multiplier, which is often set as a constant in practice [11, 21].

Objective (22) can be interpreted as comprising two key components: (1) maximization of the learned Q-function, which encourages the agent to explore unknown regions and facilitates policy improvement; and (2) a policy distribution regularization term, which enforces alignment with optimal behavior policies and thereby promotes the exploitation of high-quality actions.

Conceptual similarities exist between our method and both self-imitation learning [28] and tandem learning [29, 17]. Self-imitation learning focuses on exploiting high-reward behaviors by encouraging the policy to revisit successful past experiences, typically requiring complete trajectories and modifications to the data pipeline. In contrast, our method operates directly on individual samples from the buffer, enabling more flexible and efficient sample utilization. Tandem learning, by comparison, decomposes the learning process into active and passive agents to facilitate knowledge transfer, with a primary emphasis on value learning, whereas our approach is centered on policy extraction.

**Algorithm 1** Flow RL

**Require:** Critic $Q^{\pi_\theta}$, critic $Q^{\pi_\beta^*}$, value $V^{\pi_\beta^*}$, flow model $v_\theta$, replay buffer $\mathcal{D} = \emptyset$, weighting function $f$

1: **repeat**
2:     **for** each environment step **do**
3:         $a \sim \pi_\theta(a|s), \quad r, s' \sim P(s'|s, a)$
4:         $\mathcal{D} \leftarrow \mathcal{D} \cup \{(s, a, s', r)\}$
5:     **end for**
6:     **for** each gradient step **do**
7:         **Estimate value for** $\pi_\theta$ : Update $Q^{\pi_\theta}$ by (3),
8:         **Estimate value for** $\pi_{\beta^*}$ : Update $Q^{\pi_{\beta^*}}$ by (19), update $V^{\pi_{\beta^*}}$ by (18)
9:         Update $v_\theta$ by (22)
10:    **end for**
11: **until** reach the max environment steps

## 5 Experiments

To comprehensively evaluate the effectiveness and generality of **FlowRL**, we conduct experiments on a diverse set of challenging tasks from DMControl [40] and HumanoidBench [36]. These benchmarks encompass high-dimensional locomotion and human-like robot (Unitree H1) control tasks. Our evaluation aims to answer the following key questions:

1. How does FlowRL compare to previous online RL algorithms and existing diffusion-based online algorithms?

2. Can the algorithm still demonstrate strong performance in the absence of any explicit exploration mechanism?

3. How does the constraint affect the performance?

We compare **FlowRL** against two categories of baselines to ensure comprehensive evaluation: **(1) Model-free RL:** We consider three representative policy parameterizations: deterministic policies (TD3 [12]), Gaussian policies (SAC [13]), and diffusion-based policies (QVPO [8], the previous state-of-the-art for diffusion-based online RL). **(2) Model-based RL:** TD-MPC2 [14], a strong model-based method on these benchmarks, is included for reference only, as it is not directly comparable to model-free methods.

### 5.1 Results and Analysis

The main results are summarized in Figure 3, which shows the learning curves across tasks. **FlowRL** consistently outperforms or matches the model-free baselines on the majority of tasks, demonstrating strong generalization and robustness, especially in challenging high-dimensional (e.g., the DMC dog domain, where $s \in \mathbb{R}^{223}$ and $a \in \mathbb{R}^{38}$) and complex control settings (e.g., Unitree H1). Compared to strong model-based baselines, FlowRL achieves comparable results but is much more efficient in terms of wall-clock time. Notably, both during the training and evaluation stage, we use flow steps $N = 1$, and do not employ any sampling-based action selection used in [8, 19]. Despite the absence of any explicit exploration mechanism, FlowRL demonstrates strong results, which can be attributed to both the inherent stochasticity and exploratory capacity of the flow-based actor and the effective exploitation of advantageous actions identified by the policy constraint. These findings indicate that, while exploration facilitates the discovery of high-reward actions, the exploitation of previously identified advantageous behaviors is equally essential.

### 5.2 Ablation Studies

One of the central designs in FlowRL is the introduction of a policy constraint mechanism. This design aims to guide the policy towards optimal behavior by adaptively weighting the constraint based on the relative advantage of the optimal behavioral policy over the current policy. To rigorously assess the necessity and effectiveness of this component, we address **Q3** by conducting ablation studies in which the policy constraint is omitted from FlowRL. Experimental results in Figure 4a indicate

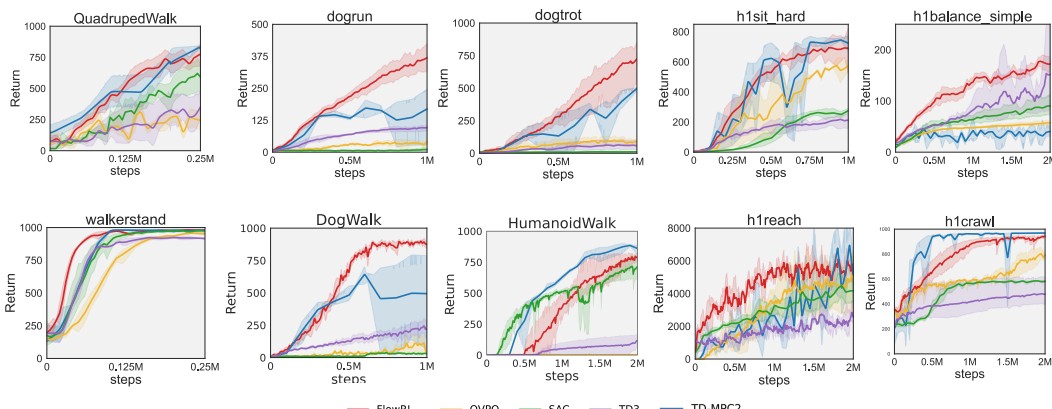

Figure 3: Main results. We provide performance comparisons for tasks (first column: DMC-easy/middle; second and third columns: DMC-hard; fourth and fifth columns: HumanoidBench). For comprehensive results, please refer to Appendix D. All model-free algorithms (FlowRL, SAC, QVPO, TD3) are evaluated with 5 random seeds, while the model-based algorithm (TD-MPC2) uses 3 seeds. Note that direct comparison between model-free methods and the model-based TD-MPC2 is not strictly fair; TD-MPC2 is included just as a reference.

that the presence of the policy constraint leads to improvements in performance and, by constraining the current policy towards the optimal behavioral policy, enhances sample efficiency. These benefits are especially pronounced in environments with complex dynamics (e.g., H1 control tasks from HumanoidBench), highlighting the importance of adaptive policy regularization in challenging task settings.

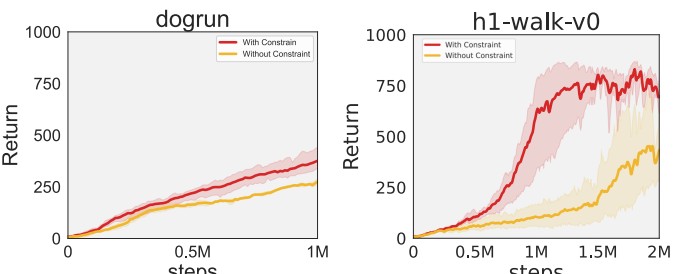
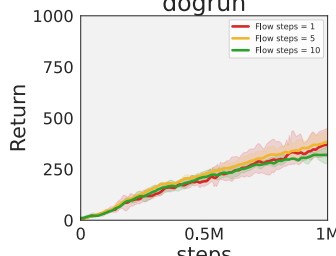

(a) Effect of the constraint: FlowRL with theconstraint achieves higher returns compared to the variant without the constraint.

(b) Sensitivity to flow steps: The number of flow steps has a limited effect on FlowRL performance.

Figure 4: Ablation studies

We also investigate the sensitivity of the algorithm to different choices of the number of flow steps (N=1,5,10). Experimental results in Figure 4b demonstrate that varying the number of flow steps has only a limited impact on the overall performance. Specifically, using a smaller number of flow steps does not substantially affect the final policy performance. On the other hand, increasing the number of flow steps results in longer backpropagation through time (BPTT) chains, which significantly increases computational complexity and training time. These findings suggest that FlowRL is robust to the choice of flow step and that single-step inference is generally sufficient for achieving stable and efficient learning in practice.We attribute this robustness to the learned Q-function acting as an implicit consistency signal (23): by assigning higher value estimates to desirable actions, the Q-function steers disparate sampling trajectories toward common high-value endpoints, so variation in integration schedule has limited impact on realized behavior. From the flow-matching perspective, the objective in Eq. (23) can thus be interpreted as comprising two complementary terms: a weighted CFM term, and a max-Q term that functions as an intrinsic, path-wise consistency constraint. Unlike the diffusion/flow-matching objectives used in generative modeling—where few-step sampling typically relies on auxiliary explicit consistency constructs (e.g., path-agreement constraints, distillation, or velocity averaging),the max-Q objective in reinforcement

learning naturally provides a form of self-consistency that aligns sampling paths toward high-return actions, obviating the need for additional consistency supervision.

$$\mathcal{L}(\theta) = \mathbb{E}_{s,a \sim \mathcal{D}, a' \sim \pi_\theta}[\underbrace{Q^{\pi_\theta}(s, a')}_{\text{Consistency}} - \lambda \left( \underbrace{f(Q^{\pi_{\beta^*}} - Q^{\pi_\theta}) \|v_\theta - (a - a^0)\|^2}_{\text{Weighted CFM}} - \epsilon \right)]. \qquad (23)$$

## 6  Conclusion

We introduces FlowRL, a practical framework that integrates flow-based generative models into online reinforcement learning through Wasserstein-2 distance constrained policy search. By parameterizing policies as state-dependent velocity fields, FlowRL leverages the expressivity of flow models to model action distributions. To align policy updates with value maximization, we propose an implicit guidance mechanism that regularizes the learned policy using high-performing actions from the replay buffer. This approach avoids explicit density estimation and reduces iterative sampling steps, achieving stable training and improved sample efficiency. Empirical results demonstrate that FlowRL achieves competitive performance.

## Acknowledgments and Disclosure of Funding

This work was jointly supported by the Joint Funds of the National Key Research and Development Program of China(No.2024YFB4711102), and National Natural Science Foundation of China (No.U22A2057) and Nanjing Major Science and Technology Special Project(No.202405017).

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

# A  Proofs in the Main Text

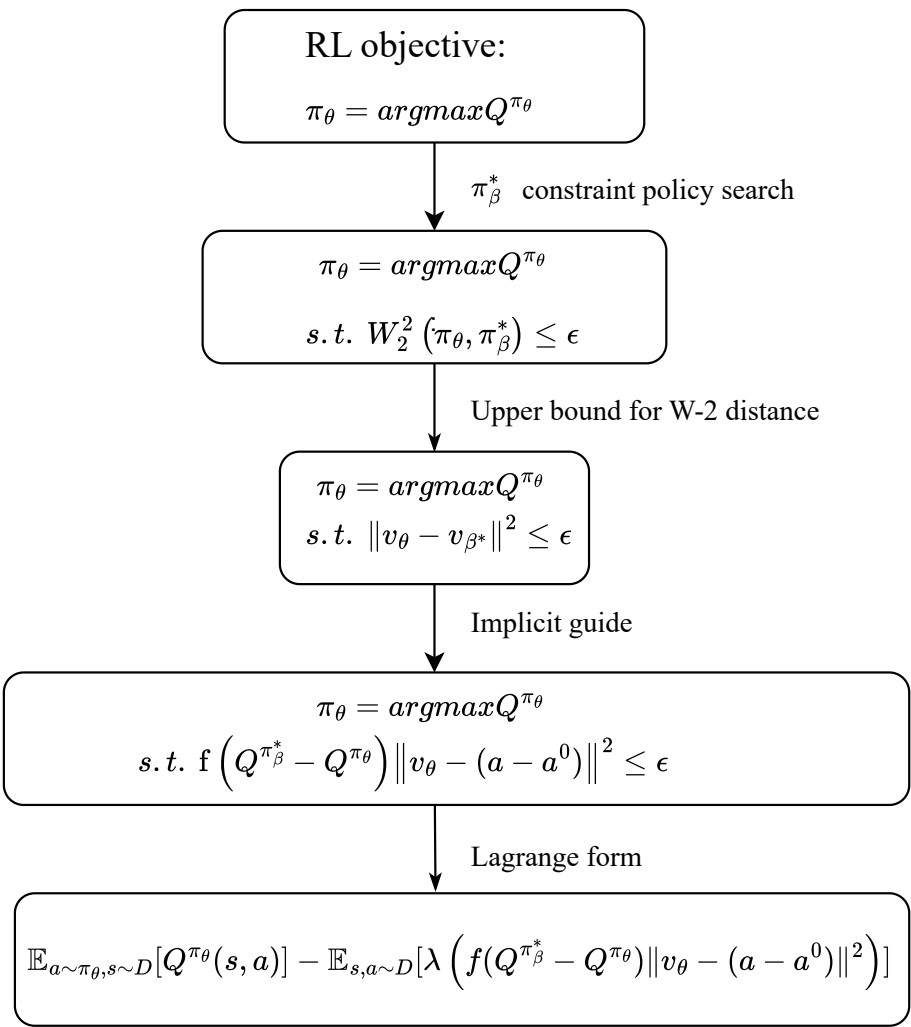

Figure 5: Theoretical sketch of FlowRL

Here, we present a sketch of theoretical analyses in Figure 5. We model the policy learning as a constrained policy search that maximizes expected returns while bounding the distance to an optimal behavior policy. To avoid sampling from $\pi_\beta^*$, we employ guided flow matching, which allows the constraint to utilize arbitrary data from the buffer. Finally, we solve the problem using Lagrangian relaxation.

## A.1  Proof for Theorem 4.1

Before the proof, we first introduce the following lemma [10]:

**Lemma 1**  :Let $\psi_1^t(x_0)$ and $\psi_2^t(x_0)$ be the two different flow maps induced by $v_1^t$ and $v_2^t$ starting from $x^0$, and assume $v_2^t$ are Lipschitz continuous in $x$ with constant $L$. Define their difference as $\Delta_t(x^0) = \psi_1^t(x^0) - \psi_2^t(x^1)$. (For notational consistency, we denote the time variable as a superscript.) Then the difference satisfies the following inequality:

$$\frac{d}{dt}\Delta_t(x_0) \leq ||v_1^t(\psi_1^t(x^0)) - v_2^t(\psi_1^t(x^0))|| + L||\Delta_t(x_0)||$$

By rewriting equivalently, we have:

$$\frac{d}{dt}\Delta_t(x^0) = \underbrace{v_1^t(\psi_1^t(x^0)) - v_2^t(\psi_1^t(x^0))}_{\delta_v(t)} + \underbrace{v_2^t(\psi_1^t(x^1)) - v_2^t(\psi_2^t(x^0))}_{\delta_\psi(t)}$$

Since $v_2^t$ is Lipschitz continuous in $x$ with constant $L$, we have:

$$\|v_2^t(x) - v_2^t(y)\| \le L\|x - y\|$$

By Lipschitz continuity,

$$\|\delta_\psi(t)\| \le L\|\Delta_t(x^0)\|$$

Then,

$$\left\|\frac{d}{dt}\Delta_t(x^0)\right\| \le \|\delta_v(t)\| + L\|\Delta_t(x^0)\|$$

This concludes the proof of the inequality satisfied by the difference of the two flow maps.

Let $v_\theta$ and $v_{\beta^*}$ be two velocity fields that induce time-evolving distributions $\pi_\theta^t(a|s)$ and $\pi_{\beta^*}^t(a|s)$, respectively((we omit the superscript $t = 1$ for policy distributions, i.e., $\pi_\theta(a|s) := \pi_\theta^1(a|s)$).). Assume $v_{\beta^*}$ is Lipschitz continuous with constant $L$. Then, define $f(t) = \|\Delta_t(x_0)\|$, by Lemma 1,we have:

$$\frac{d}{dt}f(t) \le \|\delta_v(t)\| + Lf(t),$$

where $\delta_v(t) = v_\theta(t, \psi_t^\theta(s, a^0)) - v_{\beta^*}$. Then, we have,

$$\frac{d}{dt}\left(e^{-Lt}f(t)\right) \le e^{-Lt}\|\delta_v(t)\|.$$

Then we can get (by simply intergrating from 0 to t both side and multiplying $e^{-Lt}$):

$$e^{-Lt}f(t) - f(0) \le \int_0^t e^{-Lm}\|\delta_v(m)\|dm.$$

The initial policy distribution $a^0 \sim p(a^0)$ is shared between the two velocity fields, so $f(0) = 0$. Therefore,

$$f(t) \le e^{Lt}\int_0^t e^{-Lm}\|\delta_v(m)\|dm.$$

At $t = 1$,

$$f(1) \le e^L \int_0^1 e^{-Lm}\|v_\theta(s, \psi_\theta^t(s, a^0), m) - v_{\beta^*}\|dm.$$

By taking the expectation and using Jensen's inequality:

$$\mathbb{E}_{a^0}[f(1)^2] \le e^{2L}\int_0^1 \mathbb{E}_{a\sim\pi_\theta^t}[\|v_\theta(s, a, t) - v_{\beta^*}\|^2]dt.$$

And use the definition of the Wasserstein-2 distance:

$$W_2^2(\pi_\theta, \pi_{\beta^*}) = \inf_{\gamma\in\Pi(\pi_\theta,\pi_{\beta^*})} \int_{\mathbb{R}^n\times\mathbb{R}^n} \|x - y\|^2 d\gamma(x, y),$$

where $\Pi(\pi_\theta, \pi_{\beta^*})$ denotes the set of all couplings between $\pi_\theta$ and $\pi_{\beta^*}$. Construct the following coupling $\gamma$ and define:

- $a_\theta^1 = \psi_\theta^1(x_0)$,
- $a_{\beta^*}^1 = \psi_{\beta^*}^1(x_0)$.

By definition, the coupling $\gamma$ is defined via the joint distribution of $(a \sim \pi_\theta, a \sim \pi_{\beta*})$ induced by $a_0 \sim p_0$. So, for any coupling $\gamma$,

$$W_2^2(\pi_\theta, \pi_{\beta*}) \leq \int_{\mathbb{R}^n \times \mathbb{R}^n} \|x - y\|^2 d\gamma(x, y).$$

With the constructed coupling substituted, we have

$$\int_{\mathbb{R}^n \times \mathbb{R}^n} \|x - y\|^2 d\gamma(x, y) = \mathbb{E}_{a^0}\left[\|\psi_\theta^1(a^0) - \psi_{\beta*}^1(a^0)\|^2\right] = \mathbb{E}_{a^0}[f(1)^2].$$

Recall that the flow-based policy models transport the initial distribution $p_0(a^0)$ to the final policy distributions $\pi_\theta$ and $\pi_{\beta*}$ at $t = 1$. The squared Wasserstein-2 distance between $\pi_\theta$ and $\pi_{\beta*}$ can be bounded as

$$W_2^2(\pi_\theta, \pi_{\beta*}) \leq \mathbb{E}_{a^0}[f(1)^2]. \tag{24}$$

Thus,

$$W_2^2(\pi_\theta, \pi_{\beta*}) \leq e^{2L} \int_0^1 \mathbb{E}_{a \sim \pi_\theta^t}[\|v_\theta(s, a^t) - v_{\beta*}(s, a)\|^2]ds. \tag{25}$$

## A.2 Proof for Theorem 4.2

The weighted loss can be written as:

$$\mathcal{L}_W(\theta) = \int_{s \sim D} \rho(s) \int_{s, a \sim D} f(s, a)\, \pi_k(a|s)\, \|v_\theta(s, a^t, t) - (a - a^0))\| da\, ds$$

where $\rho(s)$ is the state distribution in replay buffer, $a^0 \sim \mathcal{N}(0, I^2)$, $t \sim \mathcal{U}(0, 1)$, $a^t = ta + (1 - t)a^0$.

Assuming the weighted policy distribution is:

$$\pi_{k+1}(a'|s) = \frac{f(s, a)\, \pi_k(a|s)}{\mathcal{Z}(s)}, \quad \text{where} \quad \mathcal{Z}(s) = \int_{s, a \sim D} f(s, a)\, \pi_k(a|s)\, da.$$

Substituting above $\pi_{k+1}(a'|s)$ into the loss function, we have:

$$\mathcal{L}_W(\theta) = \int_{s \sim D} \rho(s)\, \mathcal{Z}(s) \int_{s, a \sim D} \pi_{k+1}(a'|s)\, \|v_\theta(s, a^t, t) - (a - a^0))\| da\, ds.$$

The expectation form:

$$\mathcal{L}_W(\theta) = \mathbb{E}_{s \sim \mathcal{D},\, a \sim \pi_{k+1}(a|s)}\left[\mathcal{Z}(s)\, \|v_\theta(s, a^t, t) - (a - a^0))\|\right].$$

The gradient of $\mathcal{L}_W(\theta)$ is:

$$\nabla_\theta \mathcal{L}_W(\theta) = \mathbb{E}_{s \sim \mathcal{D},\, a \sim \pi_{k+1}(a|s)}\left[\mathcal{Z}(s)\, \nabla_\theta \|v_\theta(s, a, t) - (a - a^0))\|\right].$$

$\mathcal{Z}(s)$ does not depend on $\theta$, that means, minimizing $\mathcal{L}_W(\theta)$ is equivalent to minimizing the expected loss under the new distribution $\pi_{k+1}(a|s)$, provided that our assumption holds.

# B  Hyperparameters and Experiment Settings

In this section, we provide comprehensive details regarding the implementation of FlowRL, the baseline algorithms, and the experimental environments. All experiments are conducted on a single NVIDIA H100 GPU and an Intel(R) Platinum 8480C CPU, with two tasks running in parallel on the GPU.

## B.1  Hyperparameters

The hyperparameters used in our experiments are summarized in Table 1. For the choice of the weighting function, we use $f(x) = \mathbb{I}(x) \cdot \exp(x)$, where $\mathbb{I}(x)$ is the indicator function, i.e.,

$$\mathbb{I}(x) = \begin{cases} 1, & \text{if } x > 0 \\ 0, & \text{otherwise} \end{cases}$$

For numerical stability, the $Q$ function is normalized by subtracting its mean exclusively during the computation of the weighting function.

Table 1: Hyperparameters

| | Hyperparameter | Value |
|---|---|---|
| **Hyperparameters** | Optimizer | Adam |
| | Critic learning rate | $3 \times 10^{-4}$ |
| | Actor learning rate | $3 \times 10^{-4}$ |
| | Discount factor | 0.99 |
| | Batchsize | 256 |
| | Replay buffer size | $1 \times 10^6$ |
| | Expectile factor $\tau$ | 0.9 |
| | Lagrangian multiplier $\lambda$ | 0.1 |
| | Flow steps $N$ | 1 |
| | ODE Slover | Midpoint Euler |
| **Value network** | Network hidden dim | 512 |
| | Network hidden layers | 3 |
| | Network activation function | mish |
| **Policy network** | Network hidden dim | 512 |
| | Network hidden layers | 2 |
| | Network activation function | elu |

## B.2 Baselines

In our experiments, we have implemented SAC, TD3, QVPO and TD-MPC2 using their original code bases and slightly tuned them to match our evaluation protocol to ensure a fair and consistent comparison.

- For SAC [13], we utilized the open-source PyTorch implementation, available at `https://github.com/pranz24/pytorch-soft-actor-critic`.

- TD3 [12] was integrated into our experiments through its official codebase, accessible at `https://github.com/sfujim/TD3`.

- QVPO [8] was integrated into our experiments through its official codebase, accessible at `https://https://github.com/wadx2019/qvpo`.

- TD-MPC2 [14] was employed with its official implementation from `https://github.com/nicklashansen/tdmpc2` and used their official results.

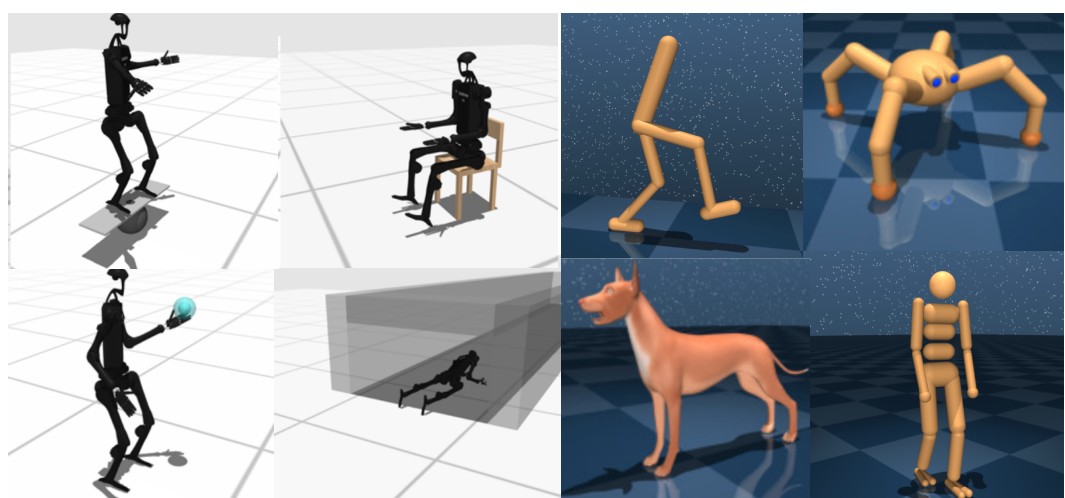

Figure 6: Task domain visualizations

## B.3 Environment Details

We validate our algorithm on the DMControl [40] and HumanoidBench [36], including the most challenging high-dimensional and Unitree H1 humanoid robot control tasks. On DMControl, tasks are categorized into DMC easy & middle (walker and quadruped domains), and DMC hard (dog and humanoid domains). On HumanoidBench, we focus on tasks that do not require dexterous hands.

| Task | State dim | Action dim |
|---|---|---|
| Walker Run | 24 | 6 |
| Walker Stand | 24 | 6 |
| Quadruped Walk | 78 | 12 |
| Humanoid Run | 67 | 24 |
| Humanoid Walk | 67 | 24 |
| Dog Run | 223 | 38 |
| Dog Trot | 223 | 38 |
| Dog Stand | 223 | 38 |
| Dog Walk | 223 | 38 |

Table 2: Task dimensions for DMControl.

| Task | Observation dim | Action dim |
|---|---|---|
| H1 Balance Hard | 77 | 19 |
| H1 Balance Simple | 64 | 19 |
| H1 Crawl | 51 | 19 |
| H1 Maze | 51 | 19 |
| H1 Reach | 57 | 19 |
| H1 Sit Hard | 64 | 19 |

Table 3: Task dimensions for HumanoidBench.

## B.4 Toy Example Setup

We consider a 2D toy example as follows. The behavior policy is a Gaussian mixture model with 10 components, each with mean

$$\mu_k = (10 \cos(2\pi k/10),\ 10 \sin(2\pi k/10)),\quad k = 0, 1, \ldots, 9,$$

and covariance $I$. The initial distribution is a Gaussian $\mathcal{N}((0,0),\ I)$. $Q^{\pi_{\beta^*}} - Q^{\pi_\theta}$ is defined as

$$\frac{1}{600}\|x - (0,\ 8.66)\|^2 - 3,$$

and $f(x) = \mathbb{I}(x) \cdot x$. Flow steps $N = 5$.

## C  Limitation and Future Work

In this work, we propose a flow-based reinforcement learning framework that leverages the behavior-optimal policy as a constraint. Although competitive performance is achieved even without explicit exploration, investigating efficient adaptive exploration mechanisms remains a promising direction for future research.

## D  More Experimental Results

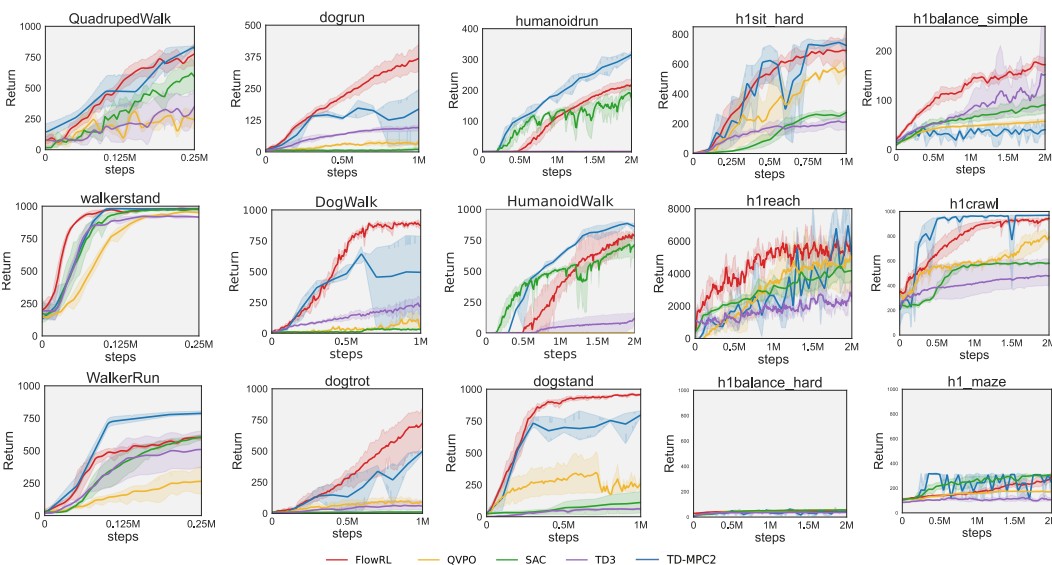

Figure 7: Experimental results are reported on 12 tasks drawn from HumanoidBench and DMC-hard, 3 tasks from DMC-easy & middle.

