# OpenReview forum: "Flow-Based Policy for Online Reinforcement Learning"
_NeurIPS.cc/2025/Conference — NeurIPS 2025 poster_

### Official Review · Reviewer_NAN6 · 2025-06-26

**Clarity:** 3
**Significance:** 3
**Originality:** 3
**Rating:** 4
**Confidence:** 4

**Summary:**

This paper proposes an algorithm that can use flow-based policies for online constrained policy search. The key idea is to regularize the discrepancy between the online policy and the best behavior policy extracted from the replay buffer, which is derived from an upper bound the the Wasserstein-2 distance between the two policies. The authors demonstrate the efficacy of this algorithm with experiments on DMControl tasks and HumanoidBench tasks, and they also show the efficacy of their proposed regularization on two tasks.

**Questions:**

1. Since Eq (13) is the key idea of the proposed algorithm, how is the variant, without constraint, performs for all the tasks? I notice that for dogrun, the performance between "with constraint" and "without constraint" is quite small, and it seems that the variant "without constraint" achieves a return of around 250, which is still higher than all other methods.

2. Why are the results of alternative methods not presented for h1-walk-v0?

3. While you argue that the proposed methods takes less wall clock time when compared to diffusion-based methods, there is no quantitative results for that. Could you provide some?

4. There is only one diffusion-based method in your selected alternative methods, so it is hard to tell if this method indeed enjoys performance benefit. Since you have mentioned several algorithms in your related work section, could you please provide results of more diffusion-based method?

5. Since the motivation of using more expressive policy representations is to model multi-modal policies, could you please provide either qualitative or quantitative result for this aspect?

I will re-consider my evaluation if these questions can be answered.

**Ethical Concerns:**

["NO or VERY MINOR ethics concerns only"]

**Final Justification:**

Although I understand that diffusion based policies can be difficult to train in online RL, the lack of comparison with alternative methods remains to be a weakness of this method. So I keep my original scoring.

**Limitations:**

There is no negative social impact.

**Paper Formatting Concerns:**

Not applicable.

**Quality:**

2

**Strengths And Weaknesses:**

# Strengths
1. This paper addresses an important topic in the recent RL literature: how to use more expressive policy classes for online RL.
2. The key idea looks reasonable, and the proposed algorithm is easy to implement.

# Weakness
1. The experiment results are not very convincing. See the question section below.
2. The flow of this paper is hard to follow. In specific, the challenges of using flow models directly as policies are not discussed in detail, so readers cannot understand the reason behind using a constrained policy search setting.

---

> ### Author Rebuttal · Authors · 2025-07-30
>
> We sincerely thank you for your valueable and constructive feedback. Your insightful questions have pushed us to significantly clarify the core contributions and experimental support of our work. We have taken your comments to heart and will incorporate substantial revisions into the manuscript to address every point you raised.
>
> ###  **Clarifying the Core Motivation**
>
> Thank you for this valuable point, which allows us to further clarify our core motivation. Our framework is designed to address a fundamental objective misalignment that arises when applying generative models to online reinforcement learning.
> This misalignment stems from the discrepancy between a generative model's objective of **static distribution-matching** (i.e., replicating a fixed dataset of good actions) and the RL objective of **dynamic value-maximization** (i.e., discovering a policy that surpasses the best behavior seen so far).
>
> Our constrained optimization provides a principled solution that allows us to directly harness the powerful modeling capacity of the flow-based policy for the RL objective. The Wasserstein-2 (W2) constraint creates a trust region around high-value behaviors, which stabilizes the policy and guides its expressive power toward maximizing the Q-function. This enables the policy to safely surpass the data, rather than merely imitating the good actions.
>
> This principled approach enables our method to deliver strong performance on challenging tasks  without sacrificing the efficiency inference, this is a capability that has been elusive for prior generative policy methods.
>
> ---
> ###  **Q1: On the Performance of the Unconstrained Variant**
>
> This is a very valuable question, and your insightful observation that the unconstrained variant still outperforms baselines on `dog-run` is absolutely correct. Our response is twofold:
>
> First, as you noted, the performance gain from the constraint is more significant on `h1-walk` than on `dog-run`. This is expected in RL research;  **no single technique will dominate across all environments** due to vast differences in dynamics and reward landscapes. Our ablation studies were designed to show this:  we specifically showcase its effect on both a task where the benefit is modest and a complex environment like `h1-walk`, where its impact is critical.
>
> Your observation, however, goes even deeper. It touches upon the  challenge that the effectiveness of a policy representation often depends on the specific dynamics and reward landscape of the environment. Our  baseline results provide a clear illustration of this dependency. For instance, when comparing performance on the `dog` domain, we found that a deterministic policy (TD3) is  more effective than a  Gaussian policy (SAC). Conversely, on the DMC `humanoid` suite, the  Gaussian policy is much more better than deterministic policy
>
> We believe your observation on `dog-run` is a clear example of this principle. We hypothesize that the flow model offers a uniquely powerful representation whose inductive biases are exceptionally well-suited to the `dog` domain's dynamics. This is supported by its strong performance across the entire  `dog` domain, where it even surpasses the powerful model-based baseline TD-MPC2.
>
> Understanding precisely which policy representations are best suited for which types of environments remains a valuable direction for future research.
>
> Therefore, your observation helps disentangle our contribution into two distinct parts:
> *   The flow model  provides a powerful  policy representation.
> *   The W2 constraint makes this powerful representation robust and stable, enabling it to work reliably across a wide range of tasks **with a single set of hyperparameters**.
>
> We thank you for this insightful question, which has allowed us to articulate the dual nature of our contribution more clearly.
>
> ### **Q2: Missing Baselines on h1-walk-v0**
>
> Thank you for catching this. You are correct that these results were missing; this was an unintentional oversight on our part. We have now run all the corresponding baseline experiments on `h1-walk-v0` to ensure a complete and fair comparison. The updated results are presented in the table below.
> | Method          | Return（Mean±std） |
> | :---- | :----------- |
> | TD-MPC2(model-based)| 822.5± 27.4 |
> | QVPO      |  570.6 ±36.2|
> | SAC     | 195.1 ± 9.2 |
> | TD3     |  511.7±40.2|
> | FlowRL |  743±37.6|
>
> ###  **Q3: Quantitative Time Comparison**
>
> We have conducted a thorough analysis and present the wall-clock training and inference times in the table below.
>
> | Method | Training Time (1M Steps) | Inference Time (per action) |
> | :--- | :--- | :--- |
> | QVPO（Pytorch） |  ~11.6 hours | ~3200 µs |
> | DACER(JAX) |  ~4.9 hours | ~740 µs |
> | FlowRL (Pytorch)  | ~8.1hours | ~110 µs|
> | FlowRL (Ours, Optimized) | ~4.52 hours | ~110 µs|
>
> 1.  **Training Efficiency:** Our baseline PyTorch implementation is  **~1.4x faster** than the pytorch implementation of QVPO.  Furthermore, by applying standard, off-the-shelf optimizations (`torch.compile` and AMP), we can reduce the training time to just **~4.5 hours**, making it competitive with even highly optimized JAX-based implentaion like DACER(JAX is 4~5x faster than pytorch).
>
> 2.  **Inference Speed:**  FlowRL is nearly **30x faster**（Pytorch） per action than the multi-step sampling required by previous methods. This is a critical advantage for any real-world deployment scenario.
>
>
> ### **Q4: Comparison with More Diffusion-Based Methods**
>
> This is a good point, and we appreciate the suggestion to include a comparison with more diffusion-based methods.
>
> In our early exploratory experiments, we did investigate both DACER (which is claimed to be SOTA in its publication) and QSM. We found they struggled to achieve meaningful performance and were often unstable on our challenging tasks, particularly on the HumanoidBench suite where they  failed to learn. We also noted these methods need **careful, task-specific hyperparameter tuning**, as mentioned in their original papers.
> Due to these  issues, we initially chose to proceed with QVPO, which proved to be a more robust baseline in our testing environment.
>
> However, to provide a more comprehensive comparison as requested, we have now conducted a dedicated set of experiments with DACER and QSM on the  DMC benchmark suite. The normalized results are summarized below.
> | Tasks | FlowRL (ours) | DACER | QSM |
> | --- | --- | --- | --- |
> | DMC-easy/medium | 83 | 64 | 72 |
> | DMC-hard | 64 | 7| 18 |
>
> Thank you for suggesting adding more baselines. We will add results on the challenging HumanoidBench tasks to the final version and provide a discussion of the baseline performance in that context.
>
>
> ### **Q5: Evidence for Multi-modal Policies**
> This is an excellent question that directly probes our core motivation.
> We wish to clarify that our research's primary focus is not on re-establishing that generative models are multi-modal—a property already explored in prior work. Instead, our contribution centers on the subsequent, critical challenge: **how to make these powerful representations stable and efficient for online policy optimization.**
>
> However, to dispel any doubt about the policy's expressive capability within our framework, we are happy to provide further evidence. Our manuscript  offers a qualitative illustration in **Figure 3 (middle)**, which shows the flow-based policy learning a multi-modal action distribution to solve the  task.
>
> To validate this quantitatively, we performed a clustering analysis on the actions generated by the learned policy in `quadruped-walk`，specifically focusing on the first dimension of the action space. Our clustering analysis on the first action dimension in `quadruped-walk` was a two-step process: 1) We used DBSCAN to discover the number of action clusters. 2) We then located the center of each discovered cluster by computing its centroid .( **We sincerely apologize that we cannot include a t-SNE which is more intuitive, as conference guidelines prohibit adding new figures during the rebuttal period**) . The analysis, summarized below, confirmed the existence of two well-separated action clusters corresponding to the  distinct modes of behavior.
>
> | Analysis Method | Clusters Found | center1 | center2 |
> |-----------------|----------------|---------|--------|
> | DBSCAN          | 2             | -0.99     | 0.99    |
>
> We are grateful for your detailed review, which is really important and valuable for us.  **We are happy to engage in further discussion if anything remains unclear.**

---

### Official Review · Reviewer_JNtZ · 2025-07-02

**Clarity:** 3
**Significance:** 3
**Originality:** 3
**Rating:** 5
**Confidence:** 3

**Summary:**

The authors propose a new method, **FlowRL**, for incorporating Flow Matching into the reinforcement learning (RL) framework using a guided expectile loss.

**Questions:**

1. Does this framework support the notion of multiple tasks within the same environment?
2. How sensitive is the method to action and observation perturbations?
3. Why is the number of steps truncated in *QuadrupedWalk*? Is this due to convergence?
4. How large is the replay buffer? Does its size change during training? Are old examples replaced?
5. What are the potential failure cases or environments where FlowRL may not perform well?
6. In Figure 4, what exactly does the "policy constraint change" refer to? Is it referring to removing Equation 13 from Equation 22?
7. Why does the method appear to be largely independent of the number of flow steps, even though flow matching typically improves with more steps? Can you hypothesize why?

**Ethical Concerns:**

["NO or VERY MINOR ethics concerns only"]

**Final Justification:**

I think that the paper suggest a new strong method for online RL and would be beneficial for community.

**Limitations:**

The authors did not discuss limitations properly in the paper

**Paper Formatting Concerns:**

No concerns

**Quality:**

3

**Strengths And Weaknesses:**

### Strengths
1. Novel approach to incorporating Flow Matching into the RL framework.
2. Strong empirical performance across tasks.
3. Inference time is superior to previous methods.
4. Demonstrates robustness with respect to the number of flow steps.

---

### Weaknesses
1. No proper discussion of limitations. For example, do known issues with diffusion models also apply here?

---

> ### Author Rebuttal · Authors · 2025-07-30
>
> We sincerely thank you for your thoughtful review and positive assessment of our work. Your questions are highly insightful and help us to clarify the contributions and nuances of our framework. We are happy to address each point in detail.
>
> Thank you for raising a  point about limitations. As iterative generative models, both flow and diffusion models face similar challenges, while diffusion models introduce more complexity of their noise scheduling mechanisms.  We will add a discussion on these limitations to the manuscript.
>
> ### **Response to Q1: Support for Multiple Tasks**
> That is an excellent question regarding the generality of our framework. We would like to clarify the precise scope of our contribution.
> Our work **focuses on improving the fundamental reinforcement learning algorithm**. We address the core mechanism of how to represent a policy and extract value from experience data in the most efficient and stable way possible.
> The challenges of multi-task learning—such as learning effective task embeddings, developing general representation learning schemes, and  scaling networks—are **orthogonal** to our contribution. FlowRL is designed to serve as the core policy learning component within such problems. Its clear separation of concerns allows our method to serve as a powerful and compatible building block for future multi-task research.
>
> ###  **Response to Q2: Sensitivity to Perturbations**
> Thank you for this question. It is important to first clarify the scope of our work to address a potential misunderstanding.
> The problem you've described, concerning sensitivity to action and observation perturbations, falls squarely within the domain of **Non-Stationary Reinforcement Learning.** Our research, in line with standard practice for foundational RL algorithms, focuses on validating our core contributions on established, stationary benchmarks. This allows for a controlled and **fair comparison against prior methods that follow the same evaluation protocol**.
> Handling such perturbations typically requires a distinct set of specialized techniques that are complementary to, and orthogonal to our core algorithm. For instance, methods in that domain often employ recurrent architectures, system identification modules, or domain randomization.
>
> ###  **Response to Q3: Truncation of Steps in Quadruped-Walk**
> Thank you for this  observation. Your question highlights an important detail about our experimental protocol.
> For the relatively simple tasks within the DMC-easy/middle suites, we adopted a uniform training length of 0.25M steps to ensure an efficient allocation of our limited computational resources. Our rationale was that within this timeframe, our method had already achieved performance comparable to the reported return of prior model-free algorithms[1] (around 800 on `quadruped-walk`).
>
> However, to provide a more definitive answer and explore the full potential of our method, we extended the experiment to 1M steps. We found that performance did not just convergence but  improved.
> The result is as follows:
> | Environment | FlowRL @ 1M Steps |
> | --- | --- |
> | quadruped-walk |  978.08 |
>
> ###  **Response to Q4: Replay Buffer Details**
> To ensure a fair comparison and highlight the contributions of our algorithm itself, we followed the most standard and widely-used configuration for off-policy RL.
> - Size and Update Mechanism: The replay buffer has a fixed size of 1 million transitions. It operates as a First-In, First-Out (FIFO) queue. Throughout training, its size remains constant. Once the buffer is full, for each new transition added, the oldest one is discarded.
>
> ###  **Response to Q5: Potential Failure Cases and Limitations**
> This is an excellent and necessary question, as every algorithm has its application boundaries.
> The primary limitations of FlowRL are intrinsically linked to the challenges inherent to off-policy learning itself. Therefore, FlowRL may not perform well in scenarios where off-policy methods are generally unsuitable. A prominent example is in large-scale Reinforcement Learning from Human Feedback (RLHF) for LLMs, where the industry has largely converged on on-policy algorithms like PPO and its variants.
>
> ###  **Response to Q6: Clarification of Figure 4**
> Yes, your understanding is perfectly correct. We will clarify this in the figure caption to make it unambiguous.
>
> ### **Response to Q7: Independence from the Number of Flow Steps**
>
> This is a very **deep and insightful question** that touches upon a core advantage of our framework.
>
> In standard flow-based generative models, generation quality typically improves with more integration steps. Achieving high-quality, few-step sampling is a key research area, with established techniques including distillation (ReFlow), using an averaged velocity field (MeanFlow), or enforcing consistency (short cut models[2]).
>
> Rather than enforcing consistency through auxiliary mechanisms, our framework **achieves consistency implicitly**.
> Diverging from pure generative modeling, where consistency is typically defined by the requirement that different sampling paths yield outputs of similar quality, we argue that in the context of reinforcement learning, this notion of consistency is naturally redefined by the agent's primary objective: maximizing cumulative return. Consequently, our requirement is that **different sampling paths should all produce actions that maximize the Q-value**.
> Q-function itself serves as this consistency condition, guiding the policy toward a desirable outcome regardless of the number of integration steps. From a generative modeling perspective, our objective function can be understood as explicitly encoding this idea:
>
> $$
> \mathcal{L} = \underbrace{Q(s,a',d)}\_{\text{consistency}} - \lambda \ \underbrace{f \cdot \left\Vert v - (a - a^0) \right\Vert^{2}}\_{\text{Weighted CFM}}
> $$
>
> It is important to clarify the notation Q(s, a', d). The number of flow steps, `d`, is not an explicit input to the Q-network. However, the action `a'` is generated from the policy $\pi_\theta$ using `d`  integration steps, making `a'` functionally dependent on  `d`  . Consequently, the Q-function is implicitly conditioned on  `d`   through its dependence on the action `a'`, our notation serves to make this crucial dependency explicit in the formulation.
>
> To further substantiate this viewpoint with empirical evidence, we adopt a training paradigm analogous to that of shortcut models[2], where the number of path integration steps is randomized during training. Specifically, during the training phase that involves action sampling (i.e., for both actor and critic updates), we randomly sampled the number of flow steps, `d`, from the range [1, 5] for each generated action. To assess the resulting policy's robustness, we then evaluated its performance under two distinct inference conditions during environment interaction: one using a fixed number of steps, and another where `d` was also randomly sampled. This experimental design allows us to decouple the training and inference conditions to verify that the policy has learned a consistent mapping that produces high-quality actions across different integration paths.
>
> | Training (actor/cirtic update) |  Env interaction |Return |
> | :--- | :--- | :--- |
> | Random `d` in {1～5} | Random `d` in {1～5} | 412.6 ± 20.1 |
> | Random `d` in {1～5} | Fixed (`d=1`) | 391.45 ± 15.8 |
> | Fixed (`d=1`) | Fixed (`d=1`) | 384.86 ± 13.76 |
>
> The results strongly support our position. The policy exhibits high performance across all conditions, even in the mismatched `Random → Fixed` scenario. This demonstrates that the policy has learned a consistent mapping, capable of producing high-quality actions irrespective of the integration path. Our framework therefore provides a unified perspective: the RL objective of **maximizing the Q-value inherently serves as the consistency condition**, enabling robust and efficient few-step generation.
>
> Thank you again for your valuable review.  We are happy to engage in further discussion if anything remains unclear.
> ### **References**
> 1] **Learning a Diffusion Model Policy from Rewards via Q-Score Matching** (ICML, 2024) by Psenka, Escontrela, Abbeel, & Ma.
>
> 2] **One Step Diffusion via Shortcut Models** (arXiv, 2024) by Frans, Hafner, Levine, & Abbeel.

---

> > ### Comment · Reviewer_JNtZ · 2025-08-02
> >
> > Thank you for your prompt response! Though I would love to discuss limitations in more detail, since your comment regarding limitations for "on-policy" algorithms is not really persuasive. Are there failure modes for the algorithm? Can I just take it out of the box and be sure that it will produce the best results for an offline RL setup? Is there anything that I should be concerned about, including sensitivity to hyperparams or architecture choice, as well as cases where I would actually prefer other off-policy algorithms?

---

> > > ### Author Response · Authors · 2025-08-04
> > > **Limitations and Future Work**
> > >
> > > Thank you for this insightful follow-up. In our effort to address all technical questions during the time-constrained rebuttal period, our initial response to this point was briefer than it deserved. We appreciate the opportunity to now offer a more thorough and detailed analysis.
> > >
> > > **1. Are there failure modes for the algorithm?**
> > >
> > > Yes. FlowRL has specific limitations related to its design and application domain. Two notable examples include:
> > > * Unsuitability for Discrete Action Spaces: Like much of previous works in RL, our framework is designed for continuous control, making it unsuitable for discrete action spaces like Atari games, text-based worlds. While we are aware of recent work on discrete flows [1], adapting our method is an interesting direction for future study.
> > >
> > > * Limited Robustness to Non-Stationary Dynamics: Consistent with the  evaluation protocol of most prior work, FlowRL assumes a stationary environment. Its performance can degrade when faced with non-stationarity, such as shifts in physical parameters (e.g., its **morphology, mass distribution, or kinematic properties**) or changes in external environmental conditions (e.g., **variable friction or gravity**). Robustly adapting to such unforeseen disturbances is an important area for future work.
> > >
> > > **2. Can I just take it out of the box and be sure that it will produce the best results for an offline RL setup?**
> > >
> > > The direct answer is **no**. We want to clarify that FlowRL is a pure **online** algorithm. Applying an algorithm designed for online RL to an offline setting requires careful handling of the core challenges of  **out-of-distribution (OOD)** and **value  overestimation**, as outlined in foundational reviews[2]. A rigorous adaptation of FlowRL for the offline domain, likely by incorporating explicit conservative mechanisms, would be necessary. This remains an interesting avenue for future study.
> > >
> > > **3. Is there anything that I should be concerned about, including sensitivity to hyperparams or architecture choice?**
> > >
> > > Yes, these are practical considerations:
> > > * **Hyperparameters:** The `λ`  requires tuning. However, our studies (and as detailed in our response to Reviewer zBLY, Q2) indicate that its effect is predictable, making it relatively straightforward to tune.
> > > * **Architecture Choice:** To isolate our algorithm's contribution, we used a standard MLP architecture. We acknowledge that recent studies have shown significant gains from more sophisticated architectures. For users focused purely on performance, we would recommend trying advanced network backbones like **SiMBA-v2** [3], potentially combined with **one-shot random pruning** techniques [10].
> > >
> > > **4. Are there cases where I would actually prefer other off-policy algorithms?**
> > >
> > > Yes, the choice depends entirely on the problem domain:
> > > * For discrete action space problems, established methods like **DQN** [4] and its variants[5] are the recommended choice. For more complex discrete tasks, **discrete SAC** [6] could also be a strong candidate.
> > > * If the system dynamics are simple or if a highly expressive policy is not required, standard algorithms like **SAC** [8] or **TD3** [9] are particularly well-suited.
> > > * If the environment is non-stationary, specialized algorithms designed for this setting, such as **OMPO** [7], would be preferable.
> > >
> > >
> > > In summary, our work on FlowRL is an exploration into the challenge of policy extraction when using iterative generative models as policies in reinforcement learning. We believe that our approachcan be complementary to the specialized mechanisms found in many of the algorithms mentioned. Indeed, your insightful questions have highlighted promising directions for future research to create more general and robust agents.
> > >
> > > Thank you again for pushing us to think more deeply about these important aspects of our work. Your feedback has been invaluable. If anything remains unclear, we would be happy to discuss further.
> > >
> > > ### References
> > >
> > > [1] Gat, I. et al. (2024). *Discrete Flow Matching*.
> > >
> > > [2] Levine, S. et al. (2020). *Offline Reinforcement Learning: Tutorial, Review, and Perspectives on Open Problems*.
> > >
> > > [3] Lee, Y. et al. (2025). *Hyperspherical Normalization for Scalable Deep Reinforcement Learning*.
> > >
> > > [4] Mnih, V. et al. (2013). *Playing Atari with Deep Reinforcement Learning*.
> > >
> > > [5] Hessel, M. et al. (2018). *Rainbow: Combining Improvements in Deep Reinforcement Learning*.
> > >
> > > [6] Christodoulou, P. (2019). *Soft Actor-Critic for Discrete Action Settings*.
> > >
> > > [7] Luo, Y. et al. (2024). *OMPO: A Unified Framework for RL under Policy and Dynamics Shifts*.
> > >
> > > [8] Haarnoja, T. et al. (2018). *Soft Actor-Critic: Off-Policy Maximum Entropy Deep Reinforcement Learning with a Stochastic Actor*.
> > >
> > > [9] Fujimoto, S. et al. (2018). *Addressing Function Approximation Error in Actor-Critic Methods*.
> > >
> > > [10] Guozheng, M. et al. (2024). *Network Sparsity Unlocks the Scaling Potential of Deep Reinforcement Learning*.

---

> > > > ### Comment · Reviewer_JNtZ · 2025-08-05
> > > >
> > > > Thank you for your response, I mistakenly wrote "offline" instead of "online" in examples above where I wanted to see algorithms limitations to existent online frameworks.

---

> ### Author Response · Authors · 2025-08-06
>
> Thank you very much for the clarification.
>
> The two examples of limitations we mentioned—adapting to discrete action spaces and handling non-stationary dynamics—apply precisely to the online setting and are important directions for our future work.
> We are happy to elaborate further if anything remains unclear.
>
> We sincerely thank you for your time and guidance.

---

### Official Review · Reviewer_zBLY · 2025-07-02

**Clarity:** 3
**Significance:** 4
**Originality:** 3
**Rating:** 5
**Confidence:** 4

**Summary:**

This paper introduces FlowRL, a novel method that integrates flow-based policy representations into online reinforcement learning. The approach uses the Wasserstein-2 distance between the current policy and a behavior-optimal policy as an implicit constraint objective. Experiments on DMC-easy/middle/hard and HumanoidBench demonstrate the method’s effectiveness and efficiency.

**Questions:**

1) How is Lipschitz continuity of $v_{\beta}$ on $a$ ensured? Does the model include regularization techniques?
2) The FlowRL optimization includes hyperparameters such as $\epsilon$ and $\lambda$. How were these values selected? Experimental justification is needed.
3) The paper emphasizes reducing computational cost but lacks quantitative comparisons with other distance metrics such as KL divergence and Euclidean distance.
4) The flow step sensitivity analysis is only conducted on the high-performing task Dogrun. It remains unclear whether adjusting flow steps impacts performance on other tasks like Dogtrot or Humanoidwalk.
5) Typos: In the annotation of Figure 4(a), line 2, “theconstraint”

**Ethical Concerns:**

["NO or VERY MINOR ethics concerns only"]

**Final Justification:**

My concerns are well addressed during the rebuttal. FlowRL provides a valuable approach to integrating highly expressive policies into RL using the Wasserstein-2 distance, and extensive experimental results demonstrate its effectiveness.

**Limitations:**

yes

**Paper Formatting Concerns:**

No formatting issues.

**Quality:**

4

**Strengths And Weaknesses:**

**Strengths**

1) This paper offers a valuable perspective on incorporating highly expressive policies into RL, which is both interesting and critical. It effectively employs the Wasserstein-2 distance as an implicit constraint, offering a more efficient alternative to the commonly used but computationally intensive KL divergence.

2) Experimental results on normalized scores and training/inference time clearly demonstrate the advantages of FlowRL.

3) The paper is well-structured and clearly written, making it easy to follow.

4) The theoretical analysis of the Wasserstein-2 distance and flow matching enhances the clarity and soundness of the method.

**Weakness**

Some claims are insufficiently supported by the experiments. Please refer to the questions for details.

---

> ### Author Rebuttal · Authors · 2025-07-30
>
> Dear Reviewer,
>
> We sincerely thank you for your positive evaluation and for your insightful questions. You have identified key areas where our claims can be further substantiated, and we appreciate the opportunity to provide a more detailed justification for our methodological choices and to supplement our experimental results.
>
> ### **Response to Q1: On the Lipschitz Continuity of the Velocity Field**
> This is an excellent theoretical question. We are happy to provide a more formal justification. The Lipschitz continuity of the velocity field $v_{\beta}$ is a well-grounded assumption that follows directly from its construction within our framework and the intrinsic properties of our network architecture.
>
> First, the velocity field $v_{\beta}$ is derived by the historical behavior of the online policy; it is constructed from the past outputs of our network $v_{\theta}$. Consequently, the functional properties of $v_{\beta}$, such as continuity, are determined by the properties of the network $v_{\theta}$ itself.
>
> Second, the network $v_{\theta}$ is intrinsically Lipschitz continuous due to its architecture. Our network is a Multi-Layer Perceptron (MLP) composed of fundamental operations (linear layers and common activations like ReLU/GeLU) that are known to be Lipschitz continuous. As the composition of Lipschitz functions is also Lipschitz continuous, the entire network is guaranteed to be a Lipschitz function over any compact domain[1]. This property is a cornerstone of modern deep learning theory, ensuring stable and well-behaved models, and is extensively relied upon in foundational literature[1,2,3].
> In summary, $v_{\beta}$ inherits its functional properties from the network architecture used to generate it. Since this architecture is intrinsically Lipschitz continuous, the assumption for $v_{\beta}$ is well-founded，and we use none of regularization technique. We will add a brief note to the methodology section to make this justification explicit.
>
> ### **Response to Q2: On Hyperparameter Sensitivity and Tuning**
> You raise a crucial point regarding hyperparameter sensitivity. A robust algorithm should not be overly dependent on fine-tuning.
> 1.  **On the Role of $\epsilon$ (A Deliberate Design Choice):** We would like to clarify a key design choice in our optimization that enhances the method's practicality. While $\epsilon$ serves as a constraint boundary in the Lagrangian formulation, our practical implementation fixes the dual variable $\lambda$. In this setup, the policy gradient is **independent of the specific value of $\epsilon$**. This was an intentional choice to **remove $\epsilon$ as a tunable hyperparameter**, simplifying the method's application. We will revise our methods section to make this important point clearer.
> 2.  **On the Sensitivity of $\lambda$:** The parameter $\lambda$ balances value maximization against W2 regularization. To quantitatively address your concern, we have conducted a new, comprehensive sensitivity analysis for $\lambda$ on the challenging `Dog-run` task.
>
> | $\lambda$ | Return (Mean ± Std) |
> | :--- | :--- |
> | 0.01 | 340.58 ± 29.14 |
> | 0.1 | **384.86 ± 13.76** |
> | 1 | 352.57 ± 15.01 |
> | 10 | 5.35 ± 0.47 |
>
> As the results demonstrate, FlowRL exhibits robust performance across a wide range of $\lambda$ (from 0.01 to 1, spanning two orders of magnitude). Performance only degrades at an extreme value ($\lambda=10$), where excessive regularization  prevents the policy from improving beyond the initial data distribution. This analysis confirms that $\lambda$ is not a critical hyperparameter and is easy to tune.
>
>
> ### **Response to Q3: Quantitative Comparison of Distance Metrics**
> This is a very practical question that directly addresses one of our method's core motivations. Our choice was a deliberate one, made for both principled and practical reasons, which becomes clear when contrasted with common alternatives.
>
> **Regarding KL Divergence:** While theoretically sound, explicitly minimizing the KL divergence in an online RL setting is computationally prohibitive. The primary bottlenecks are:
>   1. Target Distribution Cost: It requires explicitly computing or sampling from the target policy,$\pi_\beta^*$, at each update step.
>   2. Log-Probability Cost: Evaluating the exact `log_prob` of actions under a flow-based policy requires computing the log-determinant of the flow model's Jacobian. Backpropagating through this operation is extremely costly.
>   3. Sampling Overhead: Reliably estimating the KL divergence requires a significant number of sampling steps, further adding to the computational burden and introducing variance.
>
>
> To provide quantitative evidence, we ran an ablation study comparing our method to a KL-constrained baseline where the KL divergence was estimated using a 5-step process.
> First, the computational overhead of the KL-based approach is significant, as shown by the wall-clock time for 1 million environment steps:
>
> | Distance Metric | Training Time (1M steps) |
> | --- | --- |
> | W2 Upper Bound (Ours) | ~8.1h |
> | Exact KL (5-step estimation) | ~11.4h |
>
> Furthermore, the learning process for the KL-constrained agent was highly erratic, with performance oscillating significantly.This is because the high variance from the KL divergence estimation introduces substantial noise into the learning signal, corrupting the policy update and leading to unstable training.
>
> | Env Steps | 0.5M | 0.7M | 1M | 1.5M | 2M |
> | --- | --- | --- | --- | --- | --- |
> | Return (KL Agent) | 98.8 ± 12.5 | 18.5 ± 9.2 | 91.5 ± 7.8 | 146.0 ± 29.7 | 179.5 ± 7.8 |
>
> (**We sincerely apologize that we cannot provide a more intuitive learning curve visualization, as conference guidelines prohibit adding new figures during the rebuttal period**).
>
> **Regarding Euclidean Distance:** We wish to clarify a key aspect of our method: computationally, the loss we optimize is in fact a standard Euclidean (L2) distance.
> This is not an arbitrary choice. In our work, we derive a tractable upper bound on the true W2 distance, which simplifies to the Euclidean (L2) distance in the space of the policy's velocity fields. This approach provides: the computational simplicity of an L2 loss, combined with the geometric and theoretical grounding of optimal transport.
>
> ### **Response to Q4: Flow Step Sensitivity Analysis**
> Thank you for this valuable suggestion. To address your comment, we have expanded our sensitivity analysis for the number of flow steps to the **`Dog-trot`** and **`Humanoid-walk`** tasks. The results confirm that performance remains robust across a reasonable range of flow steps, reinforcing our claim of efficiency.
>
> | Steps | Dog Trot | HumanoidWalk|
> | ----: | ---------: | ----------: |
> |     1 |   764±33.78 |    797±22.85 |
> |     5 |   788± 41.36|    792±23.45 |
> |    10 |  774± 27.83 |   815±24.70 |
>
> Furthermore, we would like to highlight that this empirical finding is consistent with the underlying theory. The principle that a small number of flow steps is sufficient for effective policy generation is not task-specific and should apply broadly. For a more in-depth theoretical analysis on this point, We respectfully suggest that you refer to our discussion in response to Question 7 from Reviewer JNtZ, where we elaborate on this further.
>
> Finally, thank you for catching the typo ("theconstraint") in the caption for Figure 4(a). We will correct it. We are grateful for your detailed feedback.
>
> Thank you again for your valuable review. **We are happy to engage in further discussion if anything remains unclear.**
>
> ###  **References**
> [1] Lipschitz regularity of deep neural networks: analysis and efficient estimation (Neural Information Processing Systems, 2018) by Scaman, K., & Virmaux, A.
>
> [2] Regularisation of Neural Networks by Enforcing Lipschitz Continuity (arXiv, 2021) by Gouk et al.
>
> [3] Efficient and accurate estimation of the Lipschitz constant of deep neural networks (NeurIPS, 2019) by Fazlyab et al.

---

### Official Review · Reviewer_Mh1b · 2025-07-02

**Clarity:** 3
**Significance:** 2
**Originality:** 3
**Rating:** 4
**Confidence:** 3

**Summary:**

This paper introduces FlowRL, an RL algorithm that uses flow-based policies to generate actions via ODEs and regularizes them with a Wasserstein-2 constraint toward high-reward behaviors from the replay buffer. This avoids density estimation and aligns the policy with value-based objectives. The method is efficient, simple to implement, and shows strong results on DMControl and HumanoidBench.

**Questions:**

1. How much does FlowRL rely on getting $Q^{\pi_{\beta*}}$ right? If that estimate is noisy or off, does it hurt performance? Have you tried other ways to compute it?
2. The method works without explicit exploration, but how does it do in sparse-reward tasks? Does the built-in stochasticity go far enough?
3. Can FlowRL scale to things like vision inputs or language-conditioned tasks? Any challenges there?
4. This feels close to offline RL—Since it relies on value-weighted behavior from the replay buffer,could it work directly in that setting?

I’d consider raising my score with more insight into the above questions.

**Ethical Concerns:**

["NO or VERY MINOR ethics concerns only"]

**Final Justification:**

The rebuttal clearly addressed my concerns about novelty, theory, exploration, applicability, and Q-function reliance. I don’t see any major issues left, and I recommend acceptance. I’d like to see the clarifications and new results included in the final paper.

**Limitations:**

yes.

**Paper Formatting Concerns:**

No.

**Quality:**

3

**Strengths And Weaknesses:**

#### **Strengths**:

1. The paper is easy to read, and the method is pretty straightforward. No need for complex sampling or backprop through time, which keeps things practical.
2. Compared to diffusion-based methods, FlowRL is much faster thanks to its one-step action generation. That also makes training more stable.
3. Using Wasserstein-2 as a constraint is a nice touch—it lets the policy stay close to good behaviors without needing to compute densities.
4. The experiments are solid. It works well on tough tasks like DMControl’s Dog and HumanoidBench, and fits cleanly into standard actor-critic pipelines.

#### **Weaknesses**:

1. While the paper presents a well-executed combination of known ideas, the overall novelty feels incremental. The use of flow-based policies and W2 regularization is thoughtfully integrated, but the core principle—guiding the policy using high-value replay buffer actions—follows a familiar pattern. I’d encourage the authors to better clarify what’s fundamentally new compared to prior works like SIL, AWR, and recent flow-based methods like FQL or DACER.

2. The theoretical part (bounding W2, showing improvement via regularization) is correct but fairly standard. It mostly formalizes the intuition behind the method, rather than offering new theoretical insights.
3. There’s no explicitly designed exploration strategy, although the stochastic nature of the flow-based policy does introduce some implicit exploration. This may still be a limitation in sparse-reward or exploration-heavy settings.

---

> ### Author Rebuttal · Authors · 2025-07-30
>
> Dear Reviewer,
>
> Thank you for your thoughtful and detailed feedback on our paper. We are encouraged that you found our work to achieve solid performance on challenging tasks and that you found the method practical and easy to follow.
>
> Given your reservations about our paper’s novelty and other questions, we would like to take this opportunity to address them in detail below.
>
> ### **Our Core Contribution: A Principled Framework for Flow-based Policy Improvement(w1)**
>
> Our main contribution is **a stable and effective policy improvement framework for generative and expressive policies, based on constrained optimization**. We would like to respectfully clarify why we believe this contribution is foundational rather than incremental. Our framework is designed to **solve a fundamental conflict** in online reinforcement learning: the mismatch between a generative model's static, distribution-matching objective and the dynamic, return-maximization objective of RL. The Wasserstein-2 (W2) constraint creates a trust region around high-value behaviors, which stabilizes the expressive flow-based policy. This allows it to **safely maximize the Q-function to surpass the data, rather than merely imitating good actions**. This principled approach allows our method to achieve a combination that has been elusive for prior methods: solid performance on challenging tasks while maintaining efficient inference. The significance of this is best understood by a direct comparison to the works you mentioned:
>
> | Compared To | Core Distinction & FlowRL's Advantage |
> | :--- | :--- |
> | **AWR** | **Different theoretical foundations.** AWR is derived as the closed-form solution to a **forward-KL regularized objective** which **is a form of weighted behavioral cloning** with a simple Gaussian policy. This approach induces overly **pessimistic updates** in the online setting, as it fundamentally anchors the policy to the quality of existing data. In contrast, FlowRL's constrained optimization is explicitly designed to avoid this policy-confining pessimism with expressive policies and enable robust improvement. |
> | **SIL** | **Different conceptual, implementational, and architectural approach.** FlowRL is  motivated by resolving the core objective mismatch for expressive generative policies. To achieve this, we introduce a principled constrained optimization framework that not only stabilizes the expressive flow-based policy but operates at a much more sample-efficient **transition level**. SIL directly add an auxiliary imitation loss to A3C loss and need to **collecting complete, high-return trajectories** which is often impractical and sample-inefficient. | |
> | **DACER** | **Different role for the generative model and computational cost.** DACER uses the generative model as a simple reparameterization tool(as we discussed in related work), which leads to a reliance on computationally expensive and potentially unstable Backpropagation Through Time (BPTT). FlowRL leverages the full distribution-fitting power of flow models and  avoids BPTT. |
> | **FQL** | **Different learning paradigms and use cases.** FQL is a pure offline algorithm and need a separate policy distillation step. In contrast, FlowRL is an online method that natively achieves one-step inference by directly learning an expressive flow-based policy, completely bypassing the need for distillation. |
>
> ### **On the Role of Our Theoretical Analysis(w2)**
>
> We wish to clarify the purpose of our theoretical analysis. Its primary goal was to **provide a solid mathematical foundation** for our method's design. This foundation forges a principled connection between our framework and its implementation by proving that W2 regularization guides the policy towards high-reward distributions, which in turn informs our loss function design.
>
> ### **On Exploration and Sparse-Reward Performance（w3/q2）**
>
> Thank you for this valuable question, which gives us the opportunity to elaborate on our design choices regarding exploration.
> Our framework is **fully compatible** with existing exploration techniques (e.g., RND [2] and random action noise); we intentionally omitted them in the paper to provide a **clean and focused evaluation** of our core method's inherent capabilities.
>
> To directly address your concern about exploration, **we have conducted two new sets of empirical results.**
>
> **1. Performance in a Sparse-Reward Setting:** To directly test performance in an exploration-critical setting, we modified the Metaworld "coffee-button" task (trajectory length=200) to provide  a sparse reward (+1 for success, 0 otherwise) and evaluated FlowRL against SAC, which is renowned for its strong exploration.
>
> | Algorithm | Success Rate(sparse) (Mean± std)|
> | :--- | :--- |
> | FlowRL  |  93%± 12% |
> | SAC |  73% ± 31% |
>
> **2. Quantitative Analysis of Policy Stochasticity:** We estimated the policy entropy on the highly exploration-heavy `dog-run` task using a KNN-based estimator [1] and compared it to the target entropy of a well-tuned SAC agent.
>
> | Algorithm | Average Policy Entropy |
> | :--- | :--- |
> | FlowRL (Ours) | ~ -31.27 |
> | SAC (Target Entropy) | ~ -38 |
>
> The **higher entropy (less negative)** provides quantitative evidence that our policy is inherently stochastic and exploratory.
>
> ---
>
> ### **Question 1: How much does FlowRL rely on the Q-function estimate?**
>
> Estimation of $Q^{\pi_\beta*}$ is built upon the **principled and empirically validated technique of quantile regression**, as introduced in [3]. The stability and effectiveness of this methodology are well-established, having been corroborated by prior works [4, 5].
>
> While we start from this strong foundation, we also recognize that sensitivity to value estimation error is a common challenge for all off-policy algorithms that rely on bootstrapping, rather than a vulnerability unique to our method. To address this, our implementation is further fortified with the robust technique :Clipped Double Q-learning[6], which provides a strong safeguard against the well-documented problem of value oestimation.
>
> Furthermore, our framework is not rigidly dependent on a single estimator; substituting alternatives like Extreme Q-Learning [7] yields comparable performance. Therefore, the framework's reliance on the Q-function is built upon a foundation of proven, robust, and interchangeable components, ensuring its reliability is on par with state-of-the-art methods.
>
> ### **Question 3: Can FlowRL scale to vision or language-conditioned tasks?**
>
> This is an excellent question. Our framework is designed for broad applicability, as its **core algorithm is agnostic to the input modality**.
> *   **Orthogonal Contribution:** Our focus on state-based benchmarks was a deliberate methodological choice to isolate our core contribution, a common practice in foundational reinforcement learning research. We recognize that vision introduces distinct challenges, such as learning effective representations and data augmentation. Crucially, our contribution is orthogonal to these perception challenges.
> *   **Complementary Nature:** FlowRL can be seamlessly integrated with state-of-the-art techniques developed for these specific issues, acting as a complementary component.
> *   **Experimental Evidence:** To validate this, we ran a new experiment on a visual control task. The results below show FlowRL significantly improves a strong visual RL baseline (DrQ-v2), confirming its complementary value. To ensure a fair comparison and isolate the benefits of our policy improvement, we uniformly use a one-step TD update for both the baseline and our method.
>
> | Environment (Visual) | DrQ-v2 @1M frames | DrQ-v2 + FlowRL @1M frames |
> | :--- | :--- | :--- |
> | Quadruped-Walk | 472 ± 15.2 | **613 ± 17.3** |
>
> ### **Question 4: Could FlowRL work in the offline RL setting?**
>
> Thank you for this insightful question. We would like to clarify that FlowRL is designed as a purely **online** algorithm. The distinction lies in the primary research problem each addresses:
> *   Our work concentrates on **policy extraction** within the online loop.
> *   Offline RL need to address the **distributional shift** problem, typically by preventing Q-function overestimation for out-of-distribution (OOD) actions.
>
> While our core idea is adaptable, making it effective for the offline setting would require **integrating specific conservatism mechanisms** (e.g., from CQL [8]) to handle the OOD problem.
>
> ---
> Once again, we thank you for your valuable time and constructive feedback, which has pushed us to articulate our core contributions more clearly. We hope to have the opportunity to revise our manuscript based on your suggestions.
>
> ### **References**
> [1] *Sample estimate of the entropy of a random vector* (Problems of Information Transmission, 1987) by Kozachenko & Leonenko.
>
> [2] *Exploration by Random Network Distillation* (ICLR, 2019) by Burda et al.
>
> [3] *Offline Reinforcement Learning with Implicit Q-Learning* (ICLR, 2021) by Kostrikov et al.
>
> [4] *Seizing Serendipity: Exploiting the Value of Past Success in Off-Policy Actor-Critic* (ICML, 2024) by Ji et al.
>
> [5] *Offline-Boosted Actor-Critic: Adaptively Blending Optimal Historical Behaviors in Deep Off-Policy RL* (ICML, 2024) by Luo et al.
>
> [6] *Addressing Function Approximation Error in Actor-Critic Methods* (ICML, 2018) by Fujimoto et al.
>
> [7] *Extreme Q-Learning: MaxEnt RL without Entropy* (ICLR, 2023) by Garg et al.
>
> [8] *Conservative Q-Learning for Offline Reinforcement Learning* (NeurIPS, 2020) by Kumar et al.

---

> > ### Comment · Reviewer_Mh1b · 2025-08-09
> > **Thank you for your response**
> >
> > Thank you for the detailed rebuttal and additional experiments. Your clarifications on novelty, theoretical foundation, and exploration performance have addressed my main concerns. I hope these improvements and clarifications will be incorporated into the final version.

---

> ### Author Response · Authors · 2025-08-06
> **Invitation for  Feedback**
>
> Dear Reviewer Mh1b,
>
> Thank you for your valuable time and guidance on our paper.
>
> We are writing to sincerely invite your feedback on our author rebuttal, in which we have aimed to resolve the concerns you raised. We would be very grateful for the opportunity to ensure we have fully understood and resolved your concerns.
>
> **As the discussion period is ending soon,** any feedback would be much appreciated. If anything remains unclear, we would be happy to address any further questions or points for discussion.
>
> Best regards,
>
> The Authors

---

> ### Comment · Area_Chair_ZnXs · 2025-08-07
>
> Dear Reviewer Mh1b,
>
> This is the only negative recommendation among the reviewers. Could you please check whether the rebuttal adequately addresses your concerns? Thanks for your efforts.
>
> Best regards,
>
> Your AC

---

> ### Author Response · Authors · 2025-08-09
> **Thank you for the positive feedback on our rebuttal.**
>
> Thank you for the positive feedback on our rebuttal. We are very glad to know that we have resolved your main concerns.
> We want to confirm that **all the improvements we discussed will be carefully incorporated into the final version of the paper.**
>
> Given that your concerns have been addressed, **we would be grateful if this positive outcome could be reflected in your final evaluation**.
>
> Thank you again for your valuable time and feedback.

---

### Comment · Area_Chair_ZnXs · 2025-08-02
**Friendly Reminder: Engaging with Author Rebuttals**

Dear Reviewer,

Thank you for your time and expertise in reviewing for NeurIPS 2025. As we enter the rebuttal phase, we kindly encourage you to:

1) Read and respond to authors' rebuttals at your earliest convenience.

2) Engage constructively with authors by addressing their clarifications in the discussion thread.

3) Update your review with a "Final Justification" reflecting your considered stance post-rebuttal.

Your active participation ensures a fair and collaborative evaluation process. Please don’t hesitate to reach out if you have any questions.

With gratitude,

Your AC

---

### Decision · Program_Chairs · 2025-09-17

**Decision:**

Accept (poster)

**Comment:**

This paper proposes FlowRL, a novel framework for online reinforcement learning that integrates flow-based policy representation with Wasserstein-2-regularized optimization. The positive reviewer consensus, with two Accept scores and two Weak Accepts, supports the accept decision.